

# Storage dynamics, hydrological connectivity and flux ages in a karst catchment: conceptual modelling using stable isotopes

Zhicai Zhang[1,2,4], Xi Chen[3,1], Chris Soulsby[2], Qinbo Cheng[1,4]

[1]State Key Laboratory of Hydrology-Water Resources and Hydraulic Engineering, Hohai University, Nanjing 210098, China

[2]School of Geosciences, University of Aberdeen, Aberdeen AB24 3UF, United Kingdom

[3]Institute of Surface-Earth System Science, Tianjin University, Tianjin China

[4]College of hydrology and water resources, Hohai University, Nanjing 210098, China

*Correspondence to*: Zhicai Zhang (zhangzhicai_0@hhu.edu.cn)

**Abstract:** We integrated unique high temporal resolution hydrometric and isotope data to calibrate a catchment-scale

conceptual flow-tracer model representing the two main landscape units of hillslopes, and depressions (with fast and slow flow

systems) for cock-pit karst terrain. The model could track hourly water and isotope fluxes through each landscape unit, and

we could estimate the associated storage and water age dynamics. This inferred that the fast flow reservoir in the depression

had the smallest storage, the hillslope unit was intermediate, and the slow flow reservoir had the largest. The estimated mean

ages of the hillslope unit, fast and slow flow reservoirs were 137, 326 and 493 days, respectively. Marked seasonal variability

in hydroclimate conditions and associated water storage dynamics were the main drivers of non-stationary hydrological

connectivity between the hillslope and depression. Meanwhile, the hydrological connectivity between the slow and fast slow

reservoirs had reversible directionality, which was determined by the hydraulic head within each medium. Sinkholes can make

an important hydrological connectivity between surface water and underground stream flow in the fast reservoir after heavy

rain. New water recharges the underground stream via sinkholes, introducing younger water in the underground stream flow.

**1 Introduction**

Karst aquifers are characterized by complex heterogeneous and anisotropic hydrogeological conditions which are very

different to most other geological formations (Bakalowicz, 2005; Ford and Williams, 2013). The hydrological function of the

critical zone in cockpit karst landscapes is consequently dominated by the strong influence of this unique geomorphology and

the structure of carbonate rocks. Subsurface drainage networks in karst aquifers form mixed-flow systems that integrate flow

paths with markedly different flow velocities, ranging from low velocities in the matrix and small fractures, to very high

velocities in large fractures and conduits (which often form subterranean channel networks), with associated transitions

between states of laminar and turbulent flow (White, 2007; Worthington, 2009).

Due to the high spatial variability of the hydrodynamic properties of the karst critical zone, karst hydrological models are often

conceptual, and are generally lumped at the catchment scale (e.g. Rimmer and Salingar, 2006; Fleury et al., 2007; Jukic and

Denic-Jukic, 2009; Tritz et al., 2011, Hartmann et al., 2013; Ladouche et al., 2014). Such lumped approaches, mostly based



on linear or nonlinear relationships between storage and discharge, conceptualize the physical processes at the scale of the whole karst system. However, semi-distributed lumped models need to have hydrogeological units adequately represented, in order to relate water flow in different landscape units and model parameters that have physically meaningful concepts. In order to avoid over-parameterisation (Perrin et al., 2001; Beven, 2006), many lumped models use rather simple model structures and

focus on key karst processes deemed to be dominant at particular study sites. Consequently, appropriate conceptualization of the structure and function of karst aquifers at the catchment scale is a central challenge for successful lumped modelling. Three main types of porosities – (a) micropores, (b) small fractures, and (c) large fractures and conduits – can be intuitively identified in karst systems. However, these are usually differentiated into two major categories of the matrix flow and conduit flow in conceptual models. Accordingly, the behavior of karst spring hydrographs, is often conceptualized as a two-reservoir model

to represent the dual flow system of the karst aquifer: A low permeability "slow flow" reservoir captures the function of fractured matrix blocks of the aquifer, whilst a highly permeable "fast flow" reservoir represents the larger karst conduits (Rimmer and Hartmann, 2012; Hartmann et al, 2014; Zhang et al, 2017). Based on such a strategy, structures such as soil/epikarst, vadose zone or groundwater system are usually sub-divided into different reservoirs to simulate the hydrological response of the karst aquifer at the catchment scale. However, this kind of approach cannot disaggregate water storage and

flux dynamics within different landscape units, and may be inadequate for modelling when understanding known spatial differences in hydrogeological structure is important in terms of provisioning water supplies and understanding water quality issues (Fu et al, 2016; Zhang et al, 2013). In addition, changing hydrological connectivity between different landscape units (e.g. hillslopes and depressions) is often a key control on the non-linearity of the flow responses of karst systems, though this is usually not explicitly represented in most conceptual models.

The utility of using tracers in karst hydrology is well-established and has given insights into advection-dispersion processes, physical exchange between conduits and smaller fractures/matrix, as well as identifying relevant contaminant transport parameters (e.g. Field and Pinsky, 2000; Goldscheider et al, 2008; Kübeck et al, 2013 Kogovsek and Petric, 2014). More generally, integration of tracers into rainfall-runoff models is becoming more common in hydrology and shows promise as such tracer-aided models can provide useful learning tools in hypothesis testing regarding water and solute transport (Birkel

and Soulsby, 2015a). Indeed, McDonnell and Beven (2014) have argued that such models provide a basis for ensuring that both the celerity (i.e. the speed) of the hydrological response can be captured, along with the velocity of individual water particles (i.e. the travel times) can be captured. Moreover, they identify this as one of the fundamental challenges for contemporary hydrological modelling.

In many studies, such tracer aided models have helped resolve the celerity-velocity dichotomy (Kirchner, 2003). Such

integration has helped to understand the functional influence of heterogeneity in catchment landscapes; the importance of hydrological connectivity between different landscape units and the mixing processes that regulate solute transport and control



water ages, as well as generating runoff responses (Jencso et al., 2010; Tetzlaff et al., 2014; Soulsby et al., 2015). Whilst tracer-aided models that conceptualize the transport of tracers through the karst systems via advection-dispersion, mixing, flow partitioning through different conduits, and exchange of tracer with the matrix have been widely used (Morales et al, 2010;

Charlier et al., 2012; Mudarra et al, 2014; Dewaide et al, 2016); recent modeling advances that use natural tracers, such as stable isotopes, to track water storage, flux and age dynamics have seen limited application in karst environments. Yet estimating transit times, water ages and storage dynamics with such models can provide useful metrics to characterize the karst critical zone. Additionally, incorporating isotope tracers into such models facilitates multi-objective calibration, which provides the opportunity to improve the rigor of model evaluation, constrain parameter sets and potentially reduce uncertainty

(Birkel et al, 2015b; Ala-Aho et al, 2017a).

Hydrological connectivity, which has been simply defined as the transfer of water from one part of the landscape to another (McGuire and McDonnell, 2010; Golden et al, 2014; Soulsby et al., 2015), relates to the flux of water across the landscape in different flow paths and in so doing is affected by and affects different landscape characteristics. Hydrologic connectivity is often highly dynamic in that connections/disconnections change seasonally and episodically during rainfall events. Even

though a connection may occur between two parts of the landscape, it does not necessarily suggest that water travels between those regions over the time scale of an event (McGuire and McDonnell, 2010). Connectivity is particularly relevant in karst areas as the complex subsurface hydrogeological conditions leads to frequent changes of hydrological connectivity. The system changes through periods of connection and disconnection to create dynamic feedbacks, which in turn will influence system function. Thus, understanding hydrological connectivity changes can provide key insights into the dominant processes

governing water and solute fluxes (Lexartza-Artza and Wainwright, 2009).

A major focus for hydrological research into karst critical zone function within the South West China karst region is the 1.25km$^2$ Chenqi catchment in Guizhou province. This catchment has typical cockpit karst landscape and associated karst critical zone architecture (Zhang et al., 2017). The specific aims of the paper are: (a) to simulate flow, storage, and tracer dynamics within different landscape units using a conceptual flow-tracer model; (b) to assess the nature of hydrological

connectivity between different landscape units and its controlling factors and; (c) to estimate the effects of time-varying hydrological connectivity in karstic landscapes on non-stationary water ages.

## 2 Study catchment and data

### 2.1 Study catchment

The study catchment of Chenqi, with an area of 1.25km$^2$, is located at the Puding Karst Ecohydrological Observation Station

in Guizhou Province of southwest China (Fig. 1). It is a typical cockpit karst landscape, with surrounding conical hills separated by star shaped valleys. The catchment, which is drained by a single underground channel/conduit, can be divided into two



units: depression areas with low elevation (<1340m) and steeper hillslopes with high elevation ranging from 1340~1500m. The spatial extent of the depression and hillslopes is 0.37 and 0.88 km$^2$, respectively.

Geological strata in the basin include dolostone, thick and thin limestone, marlite and Quaternary soil profiles (see cross-

sections of A-A' and B-B' in Fig 1). Limestone formations dominate the higher elevation areas with 150-200 m thickness, which lie above an impervious marlite formation. Therefore, precipitation recharging can be perched on the impervious marlite layers that discharges at the lower areas (mostly as hillslope springs). In the hillslopes, Quaternary soils are thin (less than 30cm) and irregularly developed on carbonate rocks. The outcrops of carbonate rocks cover 10-30% of the hillslope area. In the depression, soils are thick (> 2m deep). Dominant vegetation ranges from deciduous broad-leaved forest on the upper and

middle parts of the steep hillslopes to corn and rice paddy in the lower gentle foot slopes and depressions, where soils are also thicker. The paddy fields are often flooded for the heavy rainfall in summer. Additionally, there are three sinkholes outcropped in the depression where the surface and subsurface runoff can be directly drained into underground channel for the heavy rainfall events.

The catchment is located in a region with a subtropical wet monsoon climate with mean annual temperature of 20.1°C, highest

in July and lowest in January. Annual mean precipitation is 1140 mm, almost all falling in a distinct wet season from May to September and a dry season from October to April. Average monthly humidity is high, ranging from 74% to 78%.




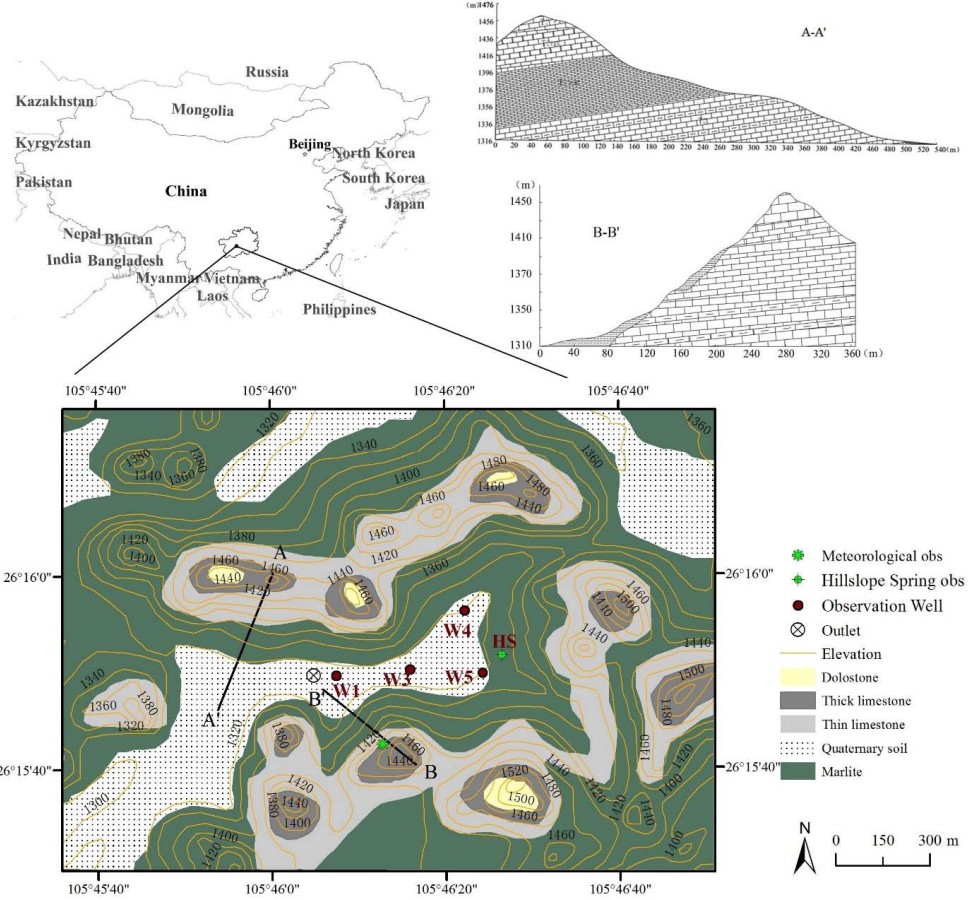

**Figure 1 Map of location, geology, geomorphology and hydrological monitoring locations in the Chenqi catchment**

### 2.2 Hydrometric and isotopic data

In Chenqi catchment, the discharge of a hillslope spring (HS) located at the foot of the eastern steep hillslope, and the

underground channel at the catchment outlet were measured with v-notch weirs (Figure 1). Their water levels were

automatically recorded by HOBO U20 water level logger (Onset Corporation, USA) with a time interval of 15 minutes.

Additionally, an automatic weather station was established on the upper hillslope to record precipitation, air temperature, wind,

radiation, air humidity and pressure. The data collection ran from 28 July 2016 to 30 October 2017. Statistical characteristics

of discharges at the spring and outlet are shown in Table 1.

**Table 1 Statistical summary of flow discharge for hillslope spring (HS) and catchment outlet (m3/s)**

| Obs | Min | Max | Range | Mean | Cv |
|---|---|---|---|---|---|
| Outlet | 0 | 0.15 | 0.15 | $4.7 \times 10^{-3}$ | 2.83 |
| HS | 0 | $1.4 \times 10^{-3}$ | $1.4 \times 10^{-3}$ | $8.5 \times 10^{-5}$ | 1.73 |





For isotope analysis, precipitation, the hillslope spring and catchment outlet flows were intensively sampled during eight rainfall events in the wet season (May to October 2017) using an autosampler set to hourly intervals (form 12 June to 14

August 2017). Groundwater in the low elevation depressions was also sampled from four wells (Figure 1), with depth below the ground surface ranging from 13 to 35m, during four rainfall events. The well screening was installed over the whole depth for each of the wells to reflect local flow exchanges at various depths in the karst. In each event, groundwater samples were collected before, during and after rainfall at each well from multiple depths.

All water samples were collected by 5 ml glass vials. The stable isotope composition of $\delta^2H$ ($\delta D$) and $\delta^{18}O$ ratios were

determined using a MAT 253 laser isotope analyser (the instrument precision ±0.5‰ for $\delta^2H$ and ±0.1‰ for $\delta^{18}O$). Isotope ratios are reported in the d-notation using the Vienna Standard Mean Ocean Water standards. Statistical characteristics of isotope signature are summarized in Table 2.

**Table 2 Statistical summary of isotope data for rainfall, hillslope spring (HS), catchment outlet and depression wells**

| Obs | δD (‰) | | | | | δ18O (‰) | | | | | lc-excess |
|---|---|---|---|---|---|---|---|---|---|---|---|
| | Max | Min | Range | Cv | Mean | Max | Min | Range | Cv | Mean | |
| Rainfall | -17.9 | -120.2 | 102.3 | 0.3 | -73.2 | 0 | -16.4 | 16.4 | 0.29 | -9.9 | -0.59 |
| Outlet | -46.9 | -73.1 | 26.2 | 0.06 | -61.9 | -5.1 | -10.6 | 5.5 | 0.09 | -8.7 | 1.13 |
| HS | -51.8 | -77 | 25.2 | 0.04 | -64.3 | -5.9 | -10.8 | 4.9 | 0.06 | -9.3 | 2.91 |
| W1 | -50.7 | -65.7 | 15 | 0.03 | -60.8 | -6.3 | -9.6 | 3.3 | 0.05 | -8.7 | 1.72 |
| W3 | -56.1 | -73.6 | 17.5 | 0.06 | -62.4 | -7.4 | -10 | 2.6 | 0.06 | -8.7 | 0.66 |
| W4 | -55 | -70.2 | 15.2 | 0.07 | -62.5 | -7.9 | -10.1 | 2.2 | 0.07 | -8.9 | 2.37 |
| W5 | -55.7 | -67.5 | 11.8 | 0.03 | -58.7 | -7.9 | -10.1 | 2.2 | 0.04 | -8.5 | 2.39 |

## 3 Methodology

### 3.1 Modeling approaches

The model used in the study was developed in previous work that used tracer data in addition to stream discharge to constrain the model structure, improve parameterization, and aid calibration (Zhang et al., 2017). The reader is referred to the original paper for full details. Briefly, this model was developed to simulate the catchment-scale water and solute transport in the dual flow system of the karst critical zone at daily time-steps. Consequently, it did not spatially disaggregate differences in flow

and tracer dynamics in different landscape units. Here we improved the model structure by separately conceptualizing the dominant hillslope and depression landscape units (Figure 2), and then use the hourly discharge and isotope time series to drive the modeling process. As Figure 2 shows, in this application, the Chenqi catchment was sub-divided into two spatially distinct units to represent the hillslope and depression and precipitation was partitioned between them. Subsequently, the depression unit was conceptualized into two flow systems, represented by "fast" and "slow" flow reservoirs which could also

exchange water. In contrast, the hillslope unit was conceptualized as just one reservoir, because of dominance of the thin soil/epikarst on water flow and lack of major aquifer influence in the steep topography (Zhang et al., 2013).




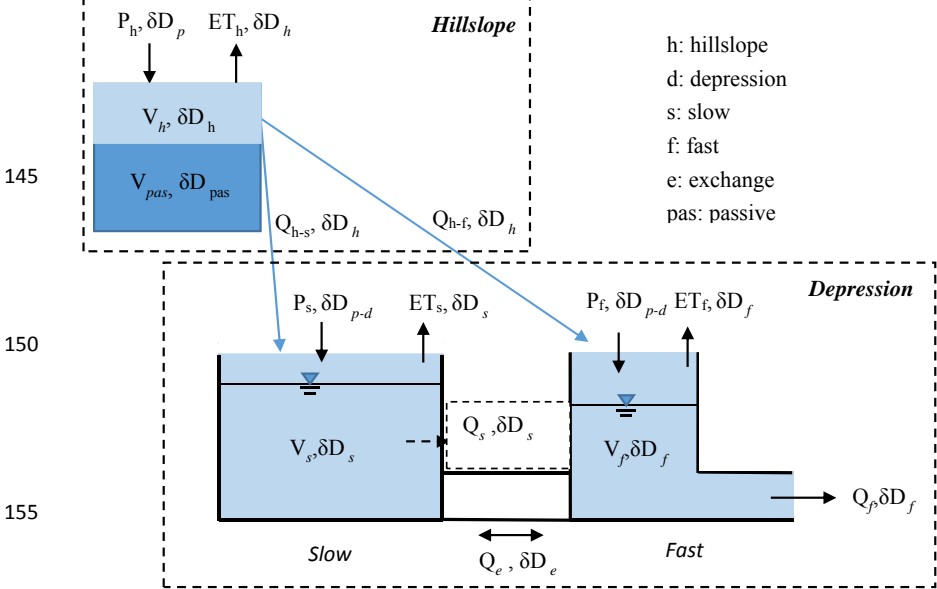

**Figure 2 Structure of the coupled flow tracer model (modified from Zhang et al., 2017). Calculation of variables and storages are presented in Appendix A.**

### 3.1.1 Hydrological simulation

The water balance in each of three reservoirs (hillslope unit, fast flow and slow flow reservoirs in depression) in the catchment are expressed as follows:

$$\frac{dV_h}{dt} = P_h - ET_h - Q_{h-s} - Q_{h-f} \tag{1}$$

$$\frac{dV_s}{dt} = P_s - ET_s + Q_{h-s} - Q_e \tag{2}$$

$$\frac{dV_f}{dt} = P_f - ET_f + Q_{h-f} + Q_e - Q_f \tag{3}$$

where $P$ is rainfall (m3 hour-1), $ET$ is evapotranspiration (m$^3$ hour$^{-1}$), $Q$ is flow discharge (m$^3$hour$^{-1}$) and $V$ is storage (m$^3$); subscripts of $h$, $s$ and $f$ represent the hillslope, slow and fast flow reservoirs, respectively, the subscripts of $h$-$s$ and $h$-$f$ represent from hillslope reservoir to slow and fast flow reservoirs, respectively, and subscript of $e$ represents flow exchange between fast and slow reservoirs. The hydrological connection and flow discharge routing for the dual flow system in the depression was derived by Zhang et al. (2017). Here, we further include the hydrological connectivity of the hillslope flow discharging into the depression reservoirs ($Q_{h-s}$ and $Q_{h-f}$ in Eqs. (1)~(3)).



### 3.1.2 Simulation of isotope ratios and estimation of water ages

The model tracks and simulates the isotope ratios for each reservoir separately, in which the isotope ratios can be complete or

partial mixed. Experimental evidence suggests that the common complete mixing assumption is overly-simplistic, in particular

for systems with pronounced switches between rapid shallow subsurface (e.g. macropores) or overland flow on the one hand

and slow matrix flow on the other hand (Van Schaik et al., 2008; Legout et al., 2009). Since the depression unit was divided

into the connected fast and slow reservoirs, a complete mixing of the isotope ratios is assumed for either reservoir. The hillslope

unit was conceptualized as a single reservoir in the new model, and complete mixing was too simplified. Hence, a partial

mixing was assumed for the hillslope (e.g. the upper active storage $V_h$ mixing with the lower passive storage $V_{pas}$ in Fig 2).

In the depression unit, the isotope mass balance in the slow and fast flow reservoirs can be expressed as:

$$\frac{di_s(V_s)}{dt} = i_{p-d}P_s - i_sET_s + i_hQ_{h-s} - i_sQ_e \tag{4}$$

$$\frac{di_f(V_f)}{dt} = i_{p-d}P_f - i_fET_f + i_hQ_{h-f} + i_sQ_e - i_fQ_f \tag{5}$$

where $i$ is the $\delta^2$H signature of the storage components (‰), the subscript of $p$-$d$ represents rainfall infiltration in depression

unit.

In the hillslope unit, only a part of the mobile water contributing to hillslope flow is mixed with water in $V_{pas}$ according to

$$\frac{di_h(V_h)}{dt} = i_pV_{p\_h} + i_{pas}V_{p\_pas} - i_hET_h - i_hQ_{h-s} - i_hQ_{h-f} + i_{pas}V_{pas\_in} - i_hV_{pas\_in} \tag{6}$$

$$\frac{di_{pas}(V_{pas})}{dt} = i_pV_{P\_pas} - i_{pas}V_{P\_pas} + i_hV_{pas\_in} - i_{pas}V_{pas\_in} \tag{7}$$

The additional volumes $V_{pas}$ (m³) is the storage of passive reservoir in hillslope which is available to determine isotope storage,

mixing, and transport in a way that does not affect the dynamics of water flux volumes. $V_{pas\_in}$ (m³) is water volume from the

active store to the passive store. $V_{p\_h}$ and $V_{p\_pas}$ (m³) are the volume of rainfall into active and passive stores, respectively.

To further quantify how catchment functioning affects water partitioning, storage and mixing, water ages are also tracked in

the model. For water age estimation in the fast and slow flow reservoirs in the depression unit, complete mixing of the inputs

is assumed and ages tracked according to determine the dynamic storage volumes on an hourly time step:

$$\frac{dAge_s(V_s)}{dt} = Age_pP_s - Age_sET_s + Age_hQ_{h-s} - Age_sQ_e \tag{8}$$

$$\frac{dAge_f(V_f)}{dt} = Age_pP_f - Age_fET_f + Age_hQ_{h-f} + Age_sQ_e - Age_fQ_f \tag{9}$$

where $Age$ is the water age.

For the age of the hillslope reservoir, the partial mixing is used:

$$\frac{dAge_h(V_h)}{dt} = Age_pV_{P\_h} + Age_{pas}V_{P\_pas} - Age_hET_h - Age_hQ_{h-s} - Age_hQ_{h-f} + Age_{pas}V_{pas\_in} - Age_hV_{pas\_in} \tag{10}$$

$$\frac{dAge_{pas}(V_{pas})}{dt} = Age_pV_{P\_pas} - Age_{pas}V_{P\_pas} + Age_hV_{pas\_in} - Age_{pas}V_{pas\_in} \tag{11}$$

where $Age_{pas}$ is passive reservoir in hillslope.



Details of the modules within the model and related equations and parameters (highlighting those calibrated) are given in Appendix A. In the equations of each module shown in Table A1, fast and slow flow reservoir storages are drained by the calibrated linear rate parameters $K_f$ and $K_s$ (hour$^{-1}$), and the exchange flow between them is calculated using the parameter $K_e$

(hour$^{-1}$) and $f$ (Table A2). Hillslope storage is drained by the exponent parameter $w$; precipitation recharging to the slow flow reservoir are calculated by the parameter $a$ (and to the fast flow reservoir by 1-$a$); Hillslope lateral flow to the slow reservoir is calculated by the parameter $b$ (to fast flow reservoir by 1-$b$); estimation of the effects of evaporative fractionation are considered by the parameter $Is$; Rainfall recharge to active and passive stores in the hillslope are calculated by the parameters $KK$ and $pp$; Exchange flow between active and passive stores in hillslope is calculated by the parameter $con$; And the weighted

isotope composition of rainfall input is calculated by the parameter $fei$. The initial range for each of the parameters was set in Table 3.

Additionally, lateral surface flow can indirectly recharge into the fast reservoir through sinkholes in the depression in heavy rainfall event. According to research at Chenqi by Peng and Wang (2012), the mean surface runoff coefficient from the hillslopes are about 10% when the hourly rainfall amount exceeds 30mm. Hence, ten percent of rainfall infiltration of hillslope

will recharge to fast flow reservoir via sinkholes in this situation (rainfall amount >30 mm/hr).

### 3.2 Modelling procedure

Modelling started on 23 July 2016, but calibration was initiated using available discharge data only from 01 November 2016. The preceding three months were used as a spin-up period (the mean of precipitation isotope signatures over the sampling period was used for the spin-up period) to fill storages, initialise storage tracer concentrations, and minimize the effects of

initial conditions on water age calculations.

The modified Kling–Gupta efficiency (KGE) criterion (Kling et al., 2012) was used as the objective function for calibration. The KGE is a three dimensional representation (Euclidean distance) of the widely used Nash–Sutcliffe criterion, overcoming some weaknesses of the latter (Schaefli and Gupta, 2007) and balancing dynamics (correlation coefficient), bias (bias ratio) and variability (variability ratio). Using flow and isotopic composition as calibration targets, objective functions were

combined to formulate a single measure of goodness of fit: KGE= (KGE$_d$ + KGE$_i$) /2 (KGE$_d$ is for discharge, KGE$_i$ is for isotopic composition).

The time series of discharge and isotope data were different in length. The high-resolution samples for stable isotope composition were collected in 8 events from 12 June to 13 August, 2017, giving a total of 589 samples. Hence, the KGE$_d$ and KGE$_i$ were each calculated using the all available data for discharge and isotope ratios, respectively. A Monte Carlo analysis

was used to explore the parameter space during calibration (Table A2). In order to derive a more behavioural parameter set, two calibration iterations were carried out. First, $10^5$ different parameter combinations within the initial ranges in Table 3 were tested. And then, the parameter ranges were reduced according to the best models (KGE >0.3) for the second calibration. This



resulted in a total of $10^5$ tested different parameter combinations. Only the best (in terms of the efficiency statistics) parameter

populations (500 parameter sets) were retained and used for further analysis, which included calculation of simulation bounds

representing posterior parameter uncertainty. The retained 500 parameter sets were applied to simulate model output, and it

was assumed that the results provide a range of behavioral models indicative of the model uncertainty in the absence of a

formal uncertainty analysis similar to Birkel et al. (2015b). Additionally, available data such as the discharge and stable isotope

signatures of hillslope spring and isotopes in the depression wells were used as qualitative "soft" data to aid model evaluation.

A regional sensitivity analysis (Freer et al., 1996) was further used to identify the most important model parameters. The

parameter sets were split into 10 groups and ranked according to the selected objective function. For each group the likelihoods

were normalized by dividing by their total, and the cumulative frequency distribution was calculated and plotted. If the model

performance is sensitive to a particular parameter there will be a large difference between the cumulative frequency

distributions compared to a 1:1 line.

### 3.3 Line-conditioned excess

The lc-excess describes the deviation of a water sample from the Local Meteoric Water Line (LMWL) in dual-isotope space,

which indicates evaporation-driven kinetic fractionation of precipitation inputs (Sprenger et al., 2016; McCutcheon et al.,

2017). With a known LMWL of $\delta^2H = a * \delta^{18}O + b$, it was thus proposed by Landwehr and Coplen (2004) that: lc-excess =

$\delta^2H - a * \delta^{18}O - b$. As oxygen has a higher atomic weight, non-equilibrium fractionation during the liquid-to-vapour phase

change will preferentially evaporate (in terms of statistical expectation) $^1H^2H^{16}O$ molecules. The isotopic signature of a water

sample affected by evaporation thus shows negative lc-excess values, and plots under the LMWL in dual-isotope space

(Landwehr et al., 2014). The LMWL of $\delta^2H = 7.77 * \delta^{18}O + 4.88$ was defined based on a daily value set of isotope signature

at precipitation from August 2016 to September 2017 in Chenqi catchment. The calculated lc-excess values were shown in

Table 2.

### 4 Results

### 4.1 Simulating flow and tracer dynamics

#### 4.4.1 The simulated flow and tracer at catchment outlet

The initial parameter ranges are listed in Table 3, along with the mean of the 500 best simulations. For the analysis, the assumed

behavioural parameter sets retained from the best 500 runs were applied to simulate model outputs with an indication of

uncertainties. The model results show that the discharge and isotope dynamics were mostly captured by the simulation ranges

at the outlet, and though some peak discharges were underestimated (Figure 3). The objective function values of KEG at the

outlet were all greater than 0.65 for the best 500 parameter sets (Table 3). As is common in coupled flow-tracer models, the



performance in the simulation of isotope was more uncertain than for discharge; $KEG_d$ ~0.8 compared with ~0.5 for $KEG_i$. In general isotope values in rainfall events depleted as the event progressed and this also depressed values in the underground stream, which the model generally reproduced (Figure 3).

The sensitive analysis results in Figure 4 show that fast flow reservoir constant ($K_f$), precipitation recharge coefficient for slow flow reservoir (*a*) and for fast flow reservoir (1- *a*), recharge coefficient of hillslope to slow flow reservoir (*b*) and to fast flow reservoir (1- *b*), coefficient for evaporation fractionation (*Is*) and weighting constant (*fei*) are generally more sensitive to the combined simulation of flow and isotopic composition.

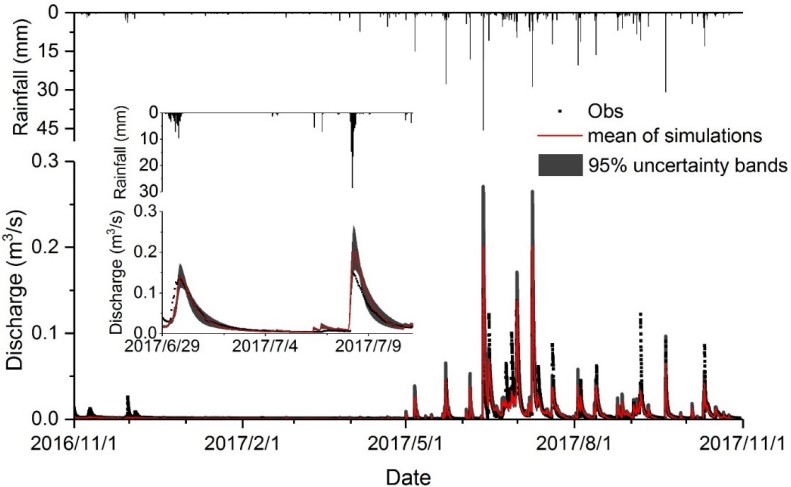

**(a) Observed and simulated stream discharge over the study period (inset shows higher resolution response over a 12 day period)**

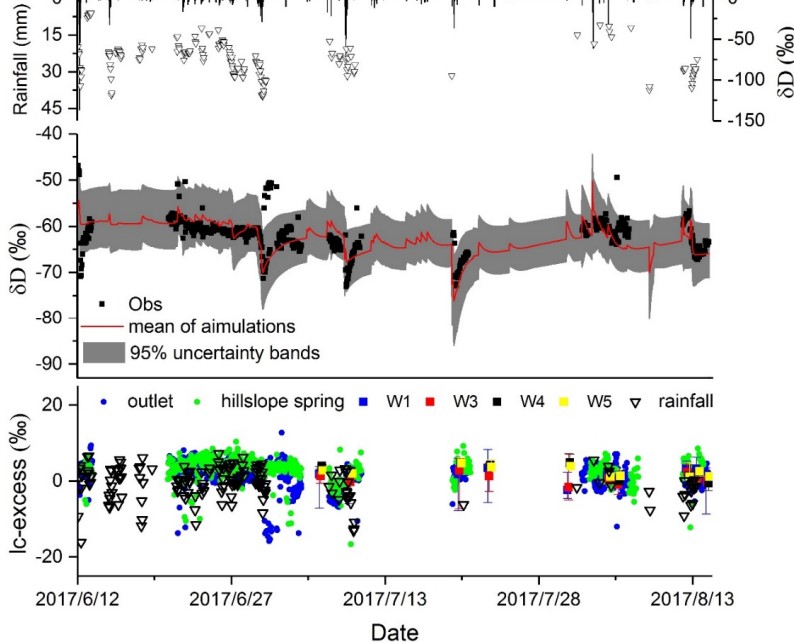

**(b) Observed and simulated deuterium, and lc-excess values**





**Figure 3 Observed stream discharge and deuterium during the study period, and discharge and deuterium simulations for the best 500 parameter sets; and lc-excess values of rainfall, outlet, hillslope spring and depression wells**

The results of isotope simulations showed that despite reproducing the general depletion, the model structure failed to capture some "rogue" high isotope values during event peaks where isotope values generally depleted (e.g. late June/early July 2017 in Figure 3). In order to explore this further, the line-conditioned excess (lc-excess) of samples was calculated from samples. The results of lc-excess values in Figure 3, with the mean of -0.59 and 1.13 at rainfall and outlet (Table 2), show a frequent evaporative fractionation effect in the underground stream over the sampling periods. Especially, there were a few samples

which showed markedly negative of lc-excess values around event peaks (e.g. 22/6, 1/7, and 5/8), indicating a strong fractionation effect. These outliers correspond to the "rogue" enriched isotope values that the model fails to capture.

The lc-excess of isotope time series of the hillslope spring and wells in the depression were estimated (Figure 3). The mean lc-excess values were also slightly negative (2.91, 1.72, 0.66, 2.37 and 2.39, respectively) for the hillslope spring, W1, W3, W4 and W5 for depression wells (Table 2), again indicating an evaporative fractionation effect on recharge water. However,

the underground stream flow (mainly reflecting the response of "fast flow" reservoirs) with the "rogue" high isotope values could not be attributed to the hillslope response or groundwater in the depression (the maximum of δD less than -50 in Table 2), because the lc-excess values of these sources were less negative than the simultaneous values of the underground stream (Figure 3) and the maximum of δD (-46.9) at outlet was larger than that at hillspring and depression wells (less than -50) in Table 2. The most likely explanation relates to flooded paddy fields which are extensively distributed in the depression during

the growing season. Consequently, large volumes of surface water are impounded in the paddy fields and exposed for evaporative fractionation. Therefore the markedly enriched isotope signals at the outlet around some event peaks would be consistent with fractionated water being displaced from the paddy fields and entering the fast flow system. This would explain the model's lack of skill in capturing such effects of evaporative fractionation.

**Table 3 Mean parameter values and fitness derived from the best 500 parameter sets after calibration**

| For Flow | $K_s$ (hour$^{-1}$) | $K_f$ (hour$^{-1}$) | $K_e$ (hour$^{-1}$) | $f$ | $a$ | $W$ | $b$ |
|---|---|---|---|---|---|---|---|
| Initial range | 40-150 | 1-40 | 800-2200 | 0.008-0.025 | 0.47-1 | 0-0.015 | 0.48-1 |
| Mean | 92 | 11 | 1549 | 0.015 | 0.68 | 0.005 | 0.54 |
| Range | 48-120 | 5-18 | 1000-2000 | 0.01-0.02 | 0.51-0.9 | 0.003-0.01 | 0.5-0.62 |

| For Isotope | $Is$ | $KK$ (×10$^4$) | $pp$ | $con$ | $fei$ | Index | Mean(range) |
|---|---|---|---|---|---|---|---|
| | | | | | | $KGE_d$ | 0.85 (0.81-0.87) |
| Initial range | 0-0.8 | 0.8-1.6 | 0-1 | 0-1 | 0.5-1 | $KGE_i$ | 0.56 (0.52-0.59) |
| Mean | 0.24 | 1.26 | 0.49 | 0.56 | 0.82 | $KGE$ | 0.7 (0.72-0.66) |
| Range | 0.002-0.6 | 1-1.5 | 0.02-0.95 | 0.04-0.97 | 0.71-0.93 | | |






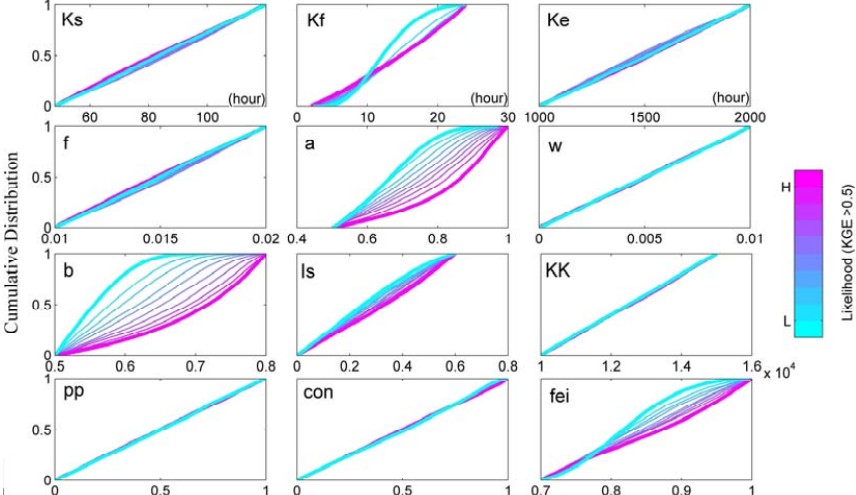

**Figure 4 Sensitivity of 12 model parameters expressed as cumulative distributions in ten levels of likelihood values for the model simulations from the lowest likelihood value (blue) to the highest likelihood value (purple). Likelihood based on KEG and rejection of values <0.5.**

**4.4.2 The simulated flow and tracer for hillslope spring and depression wells**

As a more qualitative indication of model performance, Figure 5 shows the normalized simulated discharge ($Q_n=Q_i/Q_{mean}$) of the hillslope unit had very similar seasonality and event-based dynamics to the normalized observed discharge at hillslope spring. The magnitude of the modelled discharge fluxes is, of course, different to those observed at the specific hillslope (e.g. HS at the east hillslope in Figure 1) because the simulation results represented the lumped outputs of the whole hillslope unit.

However, as a "soft" validation of the model it adds confidence that the temporal dynamics of the hillslope response are appropriately captured.

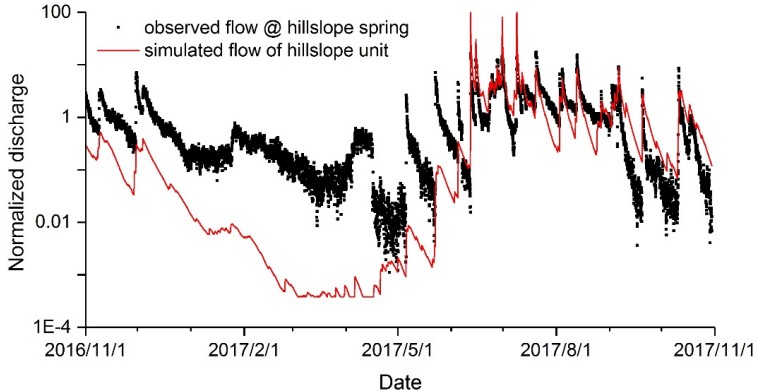

**Figure 5 Observed discharge at hillslope spring (in Fig.1) against the simulated discharge of hillslope unit (mean of the simulations for the best 500 parameter sets). Note values are normalized.**




A further qualitative test of the model was given by comparing the internal tracking of isotope dynamics in the conceptual
stores of the hillslope and slow flow reservoir with available from measured isotope values collected for the hillslope spring
and wells (Figure 6). The sampling frequency of the hillslope spring was same as at the outlet; however, there were only 10
sampling occasions from the well W1 over the dry and wet season, and the water samples were collected across a range of
depths of the well. Again, although these point measurements are not strictly comparable with the tracked isotope composition

of conceptual stores, they do give an indication of how plausible the internal states of the model are in terms of the mixing
volumes which damp the isotope inputs in precipitation. The results are encouraging, showing that the model can capture the
general directions of changes in the isotope dynamics of the hillslope spring (Figure 6a). The modelled isotope composition
of depression wells are relatively constant, however, measured values of W1 and W5 (blue and yellow points) are also
relatively constant and even samples collected at multiple depths generally have their variability bracketed within the modelled

uncertainty (Figure 6b). Recent geophysical surveys around the four wells (Chen et al, in review), show that there is relatively
low permeability rock around W1 and W5, and high permeability rock around W3 and W4. Hence, to some extent, the water
in W1 and W5 seems consistent with the slow reservoir water in depression. While the water in W3 and W4 (red and black
points in Figure 6b) are contributed by a mixing of flow from the slow and fast reservoir, especially during rainfall event (e.g.
9/7, and 20/7), leading to some samples being out of the modelled uncertainty.

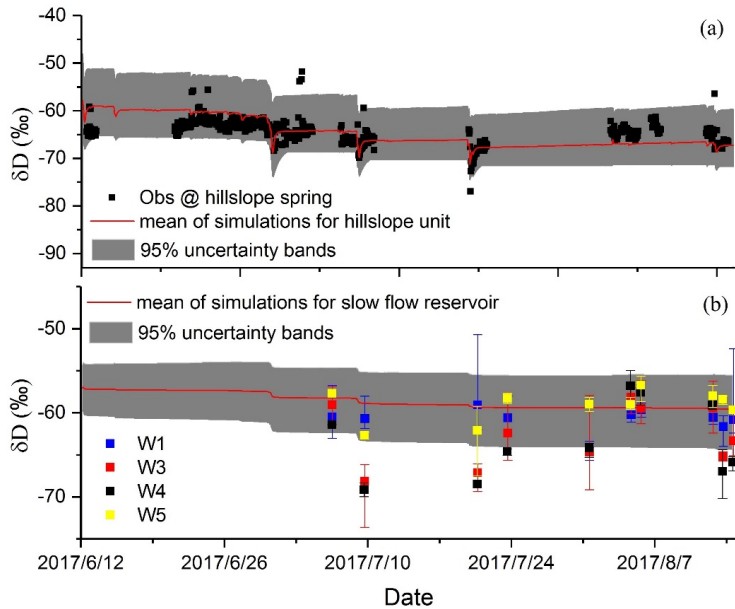


**Figure 6 Modeled isotope signature at hillslope unit and slow reservoir vs observation at hillslope spring and depression wells (the red line represents mean of the simulations for the best 500 parameter sets).**



**4.2 Storage dynamics of different reservoirs and source contributions of underground stream flow**

The storage dynamics of the catchment derived by the model in order to simulate the concurrent flow and tracer response can

be disaggregated according to conceptual stores (Figure 7). The model structure dictates that the main variability in the runoff

response to precipitation is driven by the storage dynamics, depending on hydrological connectivity between the hillslope ($V_h$),

slow ($V_s$) and fast ($V_f$) flow reservoirs (Figure 2). The modeled storage results show that slow flow reservoir was the largest

store in the catchment (~ >100mm with mean of 245 mm), consistent with the wide distribution of small fractures and matrix

pores in the karst critical zone (Zhang et al., 2011, 2017). The fast flow reservoir had the smallest storage (the mean value was

only 0.2mm) because the underground river/conduit volume represents only a very small proportion of the porosity of the

entire aquifer. Although the hillslopes cover a larger area than the depression, the thin soil, shallow epikarst and rapid drainage

resulted in a relatively small storage reservoir, with a calibrated mean value of 23mm. The discharge over the study period

showed clear seasonality, which reflects the uneven distribution of precipitation throughout the year (Figure 3). This

seasonality is mirrored somewhat differently in the storage dynamics of each reservoir (Figure 7). The storage change in fast

flow reservoir was very rapid, especially in the wet season; this reflects the rapid recharge and water release. The rapid response

of storage to rainfall was also evident in hillslope reservoir because of the low capacity and rapid response.

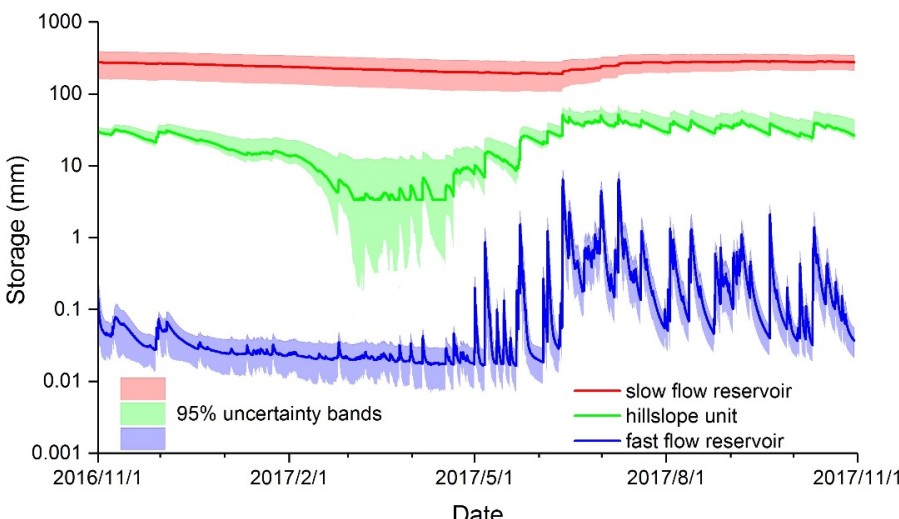

**Figure 7 Model-derived storage dynamics in hillslope ($V_h$), slow ($V_s$) and fast ($V_f$) conceptual reservoirs for the best 500 parameter sets, and the lines represent the mean of simulations**

The relative contributions of the different sources to stream flow changed with hydroclimatic conditions, and this could be

estimated using the calibrated model. Figure 8 shows that during the dry period (November 2016 to April 2017), underground

stream flow was mainly sustained from the small fractures (conceptualized as release from the slow flow reservoir). Overall,

this provided the largest proportion (78.4%) of dry season flows, followed by hillslope unit contribution (16.8%). During this

period, the rainfall infiltration contributed only limited water to the underground stream (4.8%) due to the low rainfall and



limited storage, resulting in weak hydrological connectivity between hillslope and depression. During the wet period, with the

resulting rapid increase in storage, the hillslope unit contributes much more water to the underground stream, accounting for

the largest proportion (57.5%) of overall flow, due to the strong hydrological connectivity between the hillslope and depression.

Meantime, the contribution of direct rainfall infiltration to the underground stream flow also increased (with an overall wet

season contribution of 35.6%). This likely reflects the increased influence of sinkholes and big fractures as the catchment

becomes wetter. In such conditions, during storm events, overland flow and epikarst water were collected by sinkholes and

large fractures and recharged to underground stream directly. The contribution rate of small fractures in the slow reservoir

decreased substantially (7%) although the overall magnitude of the water flux to the underground stream increased during wet

period.

The bi-directional exchange between the underground conduit and small fractures is a unique feature of the karst critical zone

(Zhang et al, 2017). During the dry period, as water table levels in the conduits drop more rapidly than in the smaller fractures,

water stored in the fractures drains into conduits and underground channels as baseflow. In the wet season, especially during

the periods of highest flow, infiltrated water quickly fills conduits where water table is higher than the adjacent fractures.

Water is temporarily stored in the conduits, and hence induces recharge in the small fractures. These bi-directional exchange

flows between the underground channels and small fractures were captured by this model (represented by fast and slow

reservoirs, respectively) and are shown in Figure 8, where the negative values represent the flux from conduits to small

fractures. This bi-directional flow was affected by the wetness conditions, being evident in the wet season and indicating both

the seasonal and short-term temporal change of hydrological connectivity between the fast and slow flow reservoirs.

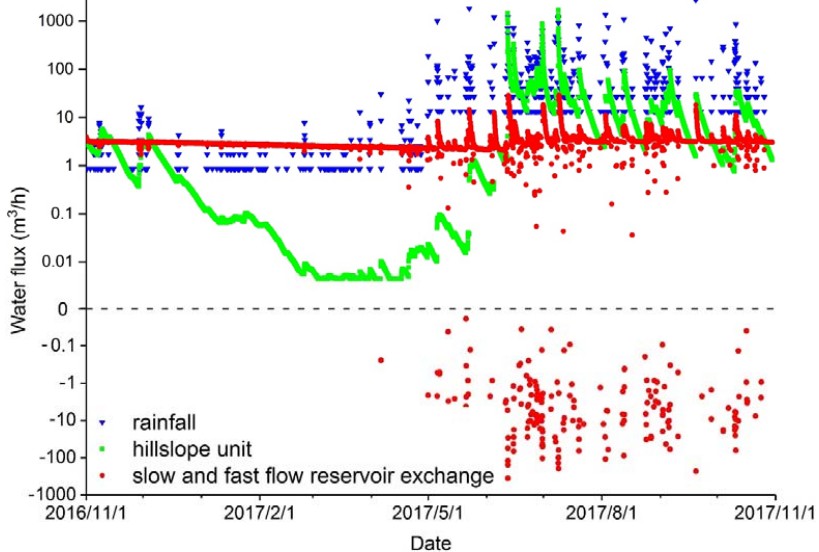

**Figure 8 Source contributions to the underground stream flow (fast reservoir) at the catchment outlet (mean of the simulations for**
**the best 500 parameter sets). The red dots above and under the dotted line represent transient reverse water fluxes from the slow**
**reservoir to fast reservoir and fast reservoir to slow reservoir, respectively.**



**4.3 Simulated flux ages from different conceptual stores**

The estimated ages of water fluxes from the different landscape units were tracked using the model. The simulated water ages were closely linked to the size of storage in each unit and the ages of the fluxes were hillslope reservoir< fast flow reservoir <

slow flow reservoir, with mean ages of 137, 326 and 493 days, respectively, and uncertainty increasing with age (Figure 9). The mean ages of water flux decreased between the dry and wet seasons: ranges from 159, 466 and 528 days for the dry season to 115, 187 and 458 days for the wet season, for the hillslope, fast flow and slow flow reservoirs respectively. The ages of fluxes from the fast flow reservoir in the underground stream generally reflected the integration of younger water fluxes from hillslope and older fluxes from the slow flow reservoir, as shown in the flux ages visualized in time series in Figure 9.

Consequently, the water age dynamics of the fast flow reservoir were relatively close to the slow flow reservoir in the dry season and close to hillslope reservoir in wet season as connectivity changed. This is consistent with the changing storage dynamics shown above. However, a distinct feature in Figure 9 is that the water ages in the fast flow reservoir were younger than which from hillslope reservoir during some events in the wet period. This again, most likely reflects the role of sinkholes in collecting water with a high proportion of new rainfall (young water) in intense wet season rain events and then recharging

underground stream rapidly due to the direct, transient connectivity.

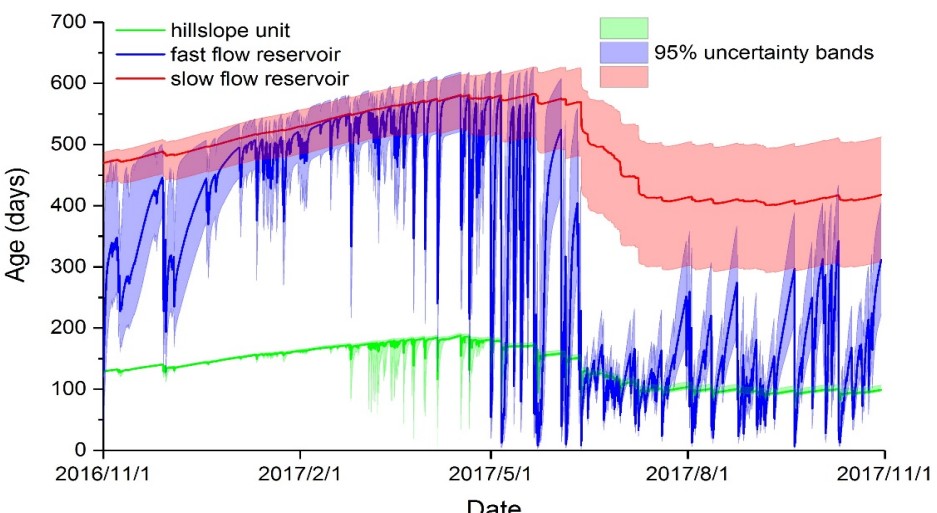

Figure 9 Mean of water flux age of hillslope, fast and slow reservoir (for the 500 best parameter sets)

The probability density functions (PDFs) of the simulated flux ages from the three reservoirs are shown in Figure 10 (using the best 500 parameter sets). The ages of fluxes from the fast flow reservoir varied from a few days to over 600 days, and it is

clearly evident that the PDF was bimodal with peaks corresponding to water ages of ~100 and ~550 days. From the water age dynamics in Figure 9, it equally clear that the bimodal distribution of ages of underground stream flow reflected the seasonality of different water sources contributions in the wet and dry seasons. The underground stream flow was dominated by older water from the matrix and small fractures during the dry period and by younger hillslope fluxes during wet period, respectively.





The water age distributions for the hillslope also showed seasonal bimodality in flux ages, albeit less pronounced, though the

model has also produced a less smooth distribution of more transient younger ages.

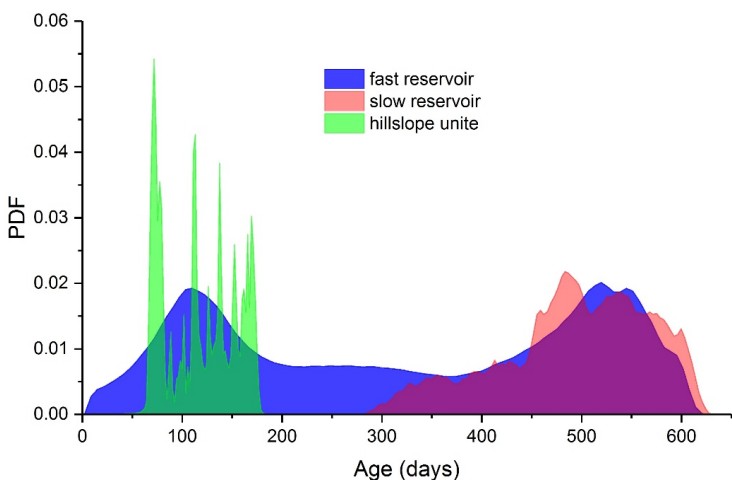

**Figure 10 Probability density functions of the simulated water ages for the best 500 runs in fluxes for all three reservoirs**

**5 Discussion**

**5.1 Using conceptual water-tracer model with low parameterization for karst catchments**

In karst areas, complex subsurface flow systems, with profound spatial heterogeneity in porosity and structure, and marked

temporal variations in hydrological connectivity, dictates that the karst critical zone is a particular challenge to hydrological

modelling. Process-based modelling is often difficult in such areas (e.g. Zhang et al, 2011; Doummar et al., 2012), because of

limited knowledge about the physical structure of subsurface drainage networks. Although conceptual modelling is more

simplistic, with relatively low parameterization, it can provide a framework for a reasonable, evidence-based first

approximation of the flow system. Hence, many conceptual models have been used in karst catchments in recent years

(Ladouche et al, 2014; Arfib and Charlier, 2016). However, high uncertainty always accompanies modelling in such complex

landscapes. The tracer-aided conceptual model used here provided an opportunity to improve the basis for model evaluation

and constrain parameter sets potentially reducing such uncertainty (Beven, 1993). Such an approach is helpful in karst regions,

because using isotope tracers as "fingerprints" means that hydrological processes can be tracked in a way that provides insights

into storage dynamics and can resolve "fast" and "slow" water fluxes and estimate their ages.

This is a significant advance on previous work, where we developed a dual-reservoir conceptual flow-tracer model for similar

conditions and calibrated the model on both flow and solute ($Ca$ and $Mg$) concentrations in three nested karst catchments

(Zhang et al, 2017). However, the earlier model was unable to differentiate the flow, storage, and tracer dynamics of different

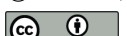



landscape units, or track water ages. The separation of the hillslope and depression in the current study improved model performance and was more informative in terms of the results showing clearly the flow and tracer dynamics within different landscape unit, as well as tracking spatially distributed storages and ages of water flux. Moreover, the model approach is potentially transferable to other cockpit karst catchments with similar landscape organization.

Although the model incorporates the effects of evaporative fractionation of water recharging the underground flow system by

a simple parameter, it can't capture some of the markedly enriched isotope signals at the outlet during the event peaks which are most likely explained by fractionated water being displaced from the paddy fields during event peaks. Hence, the skill in capturing the effects of evaporative fractionation need to be further investigated in the model; both in terms of process-based parameterisation of fractionation (e.g. Kuppel et al., 2018) and possibly differentiating paddy fields as a separate landscape unit. Though of course this would be a trade-off with increased parameterization.

**5.2 High temporal resolution isotope data for karst area**

There is a marked shift in the isotopic composition of storm event rainfall due thus weekly or even daily isotope data do not adequately capture the variability of rainfall isotope signatures at a resolution appropriate to the response times of karst systems (Coplen et al., 2008). The assessment of water ages in the critical zone is very dependent on the temporal resolution of tracer data in rainfall and stream flow for model conceptualization (Birkel et al., 2012; McDonne and Beven, 2014). The high-

frequency measurements of tracer behaviour enhanced our understanding of catchments' hydrological function and the associated time scales of hydrological response to rainfall inputs. Also, the high-resolution tracer data yielded novel insights into how the model integrates and aggregates the intrinsic complexity and heterogeneity of catchments, in order to reproduce behaviour adequately across a range of time scales (Kirchner et al., 2004). However, it should be noted that in much previous tracer-based modelling, the temporal resolution of hydrometric data (typically hourly) is at a much finer temporal resolution

than tracer data, sampled more often at daily or even weekly resolution (Stets et al., 2010; Birkel et al., 2010, 2011; McMillan et al., 2012; Soulsby et al., 2015; Ala-Aho et al., 2017a). Here, due to the marked heterogeneity of flow paths in the karst critical zone, and the very rapid (i.e. sub-daily) stream flow responses to high intensity precipitation, the modelling of flow and tracer dynamics, as well as flux age estimates, need to account for the rapid flow velocities within the karst aquifer. The response time of stream/conduit flow or groundwater level to rainfall is very short in karst catchments, e.g. typically a few

hours in small catchments like Chenqi (Zhang et al., 2013; Delbart et al., 2014; Labat and Mangin, 2015; Rathay et al., 2017). Coarser resolution data would result in increased uncertainty in the short-term components of travel times (Seeger and Weiler, 2014) and a likely bias towards longer transit times (Heidbuechel et al., 2012). Thus a significant advance in the study was that observation and model results captured the flashy (sub-daily) responses of flow and isotope signatures at hourly timescales.



### 5.3 Hydrological connectivity between different landscape units

Given the results on water storage dynamics and the relative contribution to the fast flow reservoir shown in Figures 7 and 8, it can be deduced that the storage change within each conceptual stores is the main driver of hydrological connectivity between them. During the dry period, there is weak hydrological connectivity between the hillslope and depression due to low storage. In contrast, during the wet period, hydrological connectivity between the hillslope and depression strengthens as water storage increases. In the early recession after heavy rain, large fractures in the hillslope fill, leading to large water fluxes into the

depression. Then, as storage declines, fluxes decrease and the hydrological connectivity weakens.

The hydrological connectivity and exchange between the slow and fast flow reservoirs is mainly controlled by the water level of each medium, rather than the storage. The flow directionality will change with the hydraulic gradient between the two reservoirs. Previously, few studies of hydrological connectivity have considered its directionality (e.g. Tetzlaff et al, 2007; Lexartza-Artza and Wainwright, 2009; Bracken et al., 2013; Soulsby et al., 2015). The common indices of hydrological

connectivity used are: integral connectivity scale lengths (ICSL), semivariogram-derived metrics (Ali and Roy, 2010), critical path conductivity (Knudby and Carrera, 2005), index of connectivity (Borselli et al., 2008), network index (Lane et al., 2009), relative surface connection function (Antoine et al., 2009), and saturated area (Birkel et al., 2010). Which means hydrological connectivity is usually treated as a scalar. However, the bidirectional water flux makes it fundamental to consider the directionality of connectivity within the karst critical zone. Direct hydrological connectivity between the surface and

subsurface is also important in stream flow generation in karst catchments. Besides infiltration through fractures and the matrix, concentrated infiltration from surface to underground flow systems via sinkholes is a unique aspect of transient connectivity in karst catchments. This influence is clear from the contribution of rainfall to the underground stream in Figure 8. Although this hydrological connection only occurs during heavy rain in the wet season, it is one of the most important hydrological functions of the karst critical zone. In this regard flow paths in urban areas, with transient connectivity of storm drains, have

been compared to karst (Bonneau et al. 2017) ; and whilst this gives similar short response times and a dominance of young water (Soulsby et al., 2014) urban systems are simpler and bi-directional connectivity is less significant.

### 5.4 Water ages of different conceptual stores

Through characterizing water ages of different landscape units, we can deepen our understanding of the non-linear water storage dynamics and runoff generation processes (Soulsby et al, 2015). Water ages reflect the time variance and non-linearities

of how different runoff sources are connected and the dynamics of their relative contribution to runoff generation (Birkel et al., 2012). Recent work has demonstrated the controlling effect of hydrogeological conditions on water ages in karst areas (Mueller et al, 2013). The underground stream water ages at the catchment outlet can be viewed as the time-varying integration of spatially distributed water fluxes from the hillslope unit and small fractures in the depression aquifer, which each have their own age dynamics (Figure 9). There is a distinct pattern of bimodality in the age distribution of underground stream flow

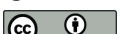



(Figure 10), which reflects the seasonality of the different water sources. Younger waters mainly come from surface water recharge through sinkholes after heavy rain and drainage from the hillslopes, whilst the low flow reservoir dominates low flows. According to the water age dynamics of different conceptual stores, it can be deduced that storage-driven changes in hydrological connectivity and associated mixing processes largely determine the nonstationary water age distribution of the underground stream. In this sense, karst catchments seem to be subject to the "inverse storage effect", where periods of high

storage facilitate release of more younger water to drainage (Hartmann, 2014).

It should be noted that the ages derived from the modelling are based on stable isotope tracers, which whilst well-suited to characterizing the influence of younger waters, are less well-suited to constraining the age of older waters (>5 years) that may be present in deeper aquifers and fine pores that contribute to the slow flow reservoir (e.g. Jasceheko et al., 2017). Thus further work is needed to assess the role of these older waters and quantify their influence on the ages of water in the older channel

(e.g. McDonnell and Beven, 2014). That said, the dominance of younger water in the outflow of responsive karst catchments is consistent with recent theoretical (Berghuijs and Kirchner (2017), larger scale (Jasecheko et al., 2016) and more local studies (Ala-aho et al., 2017b) which show that deeper, older groundwater makes limited contributions to stream flow.

**5.5 The impacts of hydrological connectivity on the pollution vulnerability**

Karst groundwater is known to be particularly sensitive to contamination, due to the aquifer structure and hydrologic behavior

(Hartmann et al 2013; Foster et al 2013). The characteristics of aquifer recharge, storage dynamics and young ages in dominant water fluxes result in a high likelihood that pollutant transport will be very rapid with only limited attenuation (Zwahlen 2003). Furthermore, during major recharge events (causing rapidly rising water tables) karst watersheds can exhibit radical changes in their groundwater flow regime and connectivity. Land management on karst hillslopes has been associated with the occurrence of diffuse pollution (Song et al., 2017). According to the results of our modelling, the hillslope is the dominant

contributor to the depression aquifer, when hydrological connectivity is high in the wet season. At the same time, any contaminants will also be rapidly delivered into the depression. Hence, it is important to note that zones of depression aquifer pollution vulnerability can be located at large distances from wells, encompassing surrounding hillslopes that can be directly connected by lateral flow paths. The resulting water age distributions show that how dominant young waters are in hillslope fluxes and the fast flow reservoir during the wet season. These can also be responsible for introducing pollution into the

underlying groundwater aquifer. The bi-directional exchange between fast and slow flow reservoirs potentially results in the more matrix pollution in the slow reservoir, in the wet season. Consequently, the attenuation and recovery of matrix pollution is very slow because of the low hydraulic conductivity. It is critical that the influence of sinkholes should be taken into account when assessing pollution sensitivity in karst areas. It can be seen from the results of water ages in Figure 9 and 10 that sinkholes facilitate the strong connectivity between the surface and ground water after heavy rain events. This potentially leads to more



pollutants entering the groundwater system and then being dispersed with the underground flows. These sensitivities need to

be recognized in restoration management of karst areas.

**6 Conclusions**

We extended a catchment-scale flow-tracer model for karst systems developed by Zhang, et al (2017) by conceptualizing two

main hydrological response units: hillslope and depression each containing fast and slow flow reservoirs. With this framework,

we could calibrate the model using high temporal resolution hydrometric and isotopic data to track hourly water and isotope

fluxes through a 1.25 km$^2$ karst catchment in southwest China. The model captured the flow and tracer dynamics within each

landscape unit quite well and we could estimate the storage, fluxes and age of water within each landscape unit. This inferred

that the fast flow reservoir had the smallest storage, the hillslope unit was intermediate, and the slow flow reservoir had the

largest. The estimated mean ages of the hillslope unit, fast and slow flow reservoirs were 137, 326 and 493 days, respectively.

Marked seasonal variability in hydroclimate and associated water storage dynamics were the main drivers of non-stationary

hydrological connectivity between the hillslope and depression. Meanwhile, the hydrological connectivity between the slow

and fast slow reservoirs had variable directionality, which was determined by the hydraulic head within each medium.

Sinkholes can make an important hydrological connectivity between surface water and underground stream flow after heavy

rain. New water recharges the underground stream via sinkholes, introducing younger water in the underground stream flow.

Whist the model here needs further development (e.g. the parameterization of isotopic fractionation in the paddy fields)

assessment and model validation requires longer and more detailed (e.g. better characterization of older waters) observation

data, it is an encouraging step forward in tracer-aided modelling of karst catchments.

*Data availability.* The isotope data as well as rainfall and flow measurements used for this paper are not publicly accessible

due to the constraints of governmental policy in China. The data were obtained through a purchasing agreement for this study.

GIS data in this study are available.

*Author contributions.* ZZ, XC, and CS conducted the modelling work and data interpretation, ZZ and QC conducted the field

and laboratory work, ZZ prepared the manuscript with contributions from co-authors.


*Competing interests.* The authors declare that they have no conflict of interest.

***Acknowledgments.*** This research was supported by The UK-China Critical Zone Observatory (CZO) Programme
(41571130071), the National Natural Scientific Foundation of China (41571020), National 973 Program of China
(2015CB452701), the National Key Research and development Program of China (2016YFC0502602), the Fundamental





Research Funds for the Central Universities (2016B04814) and the UK Natural Environment Research Council
(NE/N007468/1). In addition, we thank Sylvain Kuppel for his constructive comments.

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

**Appendix A**

**Table A1 Algorithms in this model involving hydrological and isotopic modules. Note that the calibrated parameters are shown in red at the first appearance.**

| | |
|---|---|
| $P_h = P * Area_h/Area$ <br> $P_s = (P-P_h) *a$ <br> $P_f = (P-P_h) * (1-a)$ | $Area_h$ and Area represent the hillslope and catchment area, respectively. Identify the areas using GIS. |
| $ET = \eta * ETP$ <br> $ET_h = ET * Area_h/Area$ <br> $ET_f = ET * (1- ET_h)*a$ <br> $ET_s = ET * (1- ET_h) * (1-a)$ | Potential evapotranspiration, ETP was calculated by Penman formula. Conversion factor $\eta$ was from other study in this region. |
| $Q_{h,t} = w*exp(V_{h,t}/5000)$ <br> $Q_{h-f, t} = b*Q_{h, t}$ <br> $Q_{h-s, t} = (1-b) *Q_{h, t}$ <br> $V_{h,t} = V_{h,t-1} + P_{h,t} - ET_{h,t} - Q_{h-f, t} - Q_{h-s, t}$ | Porosity decreases with increasing depth <br> Subscript t represents time |
| $\delta D_{h,t} = (\delta D_{h,t-1} * V_{h,t-1} * (1-con) + \delta D_{pas,t-1}* V_{h,t-1} * con + \delta D_{p,t} * P_{h,t} * Par_t + \delta D_{pas,t}*P_{h,t} * (1-Par_t) - \delta D_{h,t-1} * ET_{h,t} - \delta D_{h,t-1}* (Q_{h-f, t} + Q_{h-s, t}))/ V_{h,t}$ | Useing the dynamic partial mixing method (Hrachowitz et al 2013) to calculate the isotope at hillslope |
| $\delta D_{pas,t} = (\delta D_{pas,t-1} * V_{pas,t-1} + \delta D_{p,t} * P_{h,t} * (1-Par_t) + \delta D_{h,t-1}* V_{h,t-1}* (1-con) - \delta D_{pas,t-1}* V_{h,t-1} * (1-con) – (1+ls)*\delta D_{pas,t-1} * ET_{h,t}) / (V_{pas,t} + P_{h,t} * (1-Par_t))$ <br> $Par_t = pp* exp(V_{h,t}/kk)$ | |
| $Age_{h,t} = (Age_{h,t-1}+1)* V_{h,t-1} * (1-con) + (Age_{pas,t-1}+1)* V_{h,t-1}* con + Age_{p,t} * P_{h,t} * Par_t + Age_{pas,t}*P_{h,t} * (1-Par_t) – (Age_{h,t-1}+1)*ET_{h,t} - (Age_{h,t-1}+1)*(Q_{h-f, t} + Q_{h-s, t}))/ V_{h,t}$ | The Age of rainfall, $Age_{p,t}$, equal to 0. |
| $Age_{pas,t} = (Age_{pas,t-1}+1) * V_{pas,t} + Age_{p,t} * P_{h,t} * (1-Par_t) + (Age_{h,t-1}+1)* V_{h,t-1} * (1-con) - (Age_{pas,t-1}+1)* V_{h,t-1}$ | |





| | |
|---|---|
| $* (1-con) – (1+I_s)* (Age_{pas,t-1}+1) * ET_{h,t}) / (V_{pas,t} + P_{h,t}$ $* (1-Par_t))$ | |
| $Q_{s,t}=\Phi_{s,1}(I_{s,t}+Q_{h-s,t})+\Phi_{s,2}\Phi_{s,1}(I_{s,t-1}+Q_{h-s,t-1})+\Phi^2_{s,2}\Phi_{s,1}(I_{s,t-2}+Q_{h-s,t-2}) + \Phi^2_{s,2}Q_{s,t-2} + \Phi_{s,3}Q_{f,t} + \Phi_{s,2}\Phi_{s,3}Q_{f,t-1} + \Phi^2_{s,2}\Phi_{s,3}Q_{f,t-2} -\Phi_{s,1}ET_{s,t}$ | |
| $Q_{f,t} =\Phi_{f,1}(I_{f,t}+Q_{h-f,t}) + \Phi_{f,2}Q_{f,t-1} +\Phi_{f,3}(\Phi_{s,1}I_{s,t}+\Phi_{s,2}\Phi_{s,1}I_{s,t-1} +\Phi^2_{s,2}\Phi_{s,1}I_{s,t-2} +\Phi^2_{s,2}Q_{s,t-2} +\Phi_{s,3}Q_{f,t} +\Phi_{s,2}\Phi_{s,3}Q_{f,t-1} + \Phi^2_{s,2}\Phi_{s,3}Q_{f,t-2})- \Phi_{f,1} ET_{f,t}$ | |
| $\delta D_{p-d} = fei * \delta D_p$ $\delta D_{s,t}=\{\Phi_{s,1}(\delta D_{p-d,t}I_{s,t}+\delta D_{h,t}Q_{h-s,t})+\Phi_{s,2}\Phi_{s,1}(\delta D_{p-d,t-1}I_{s,t-1}+\delta D_{h,t-1}Q_{h-s,t-1})+\Phi^2_{s,2}\Phi_{s,1}(\delta D_{p-d,t-2}I_{s,t-2} +\delta D_{h,t-2}Q_{h-s,t-2}) + \Phi^2_{s,2}\delta D_{s,t-2}Q_{s,t-2} + \Phi_{s,3}\delta D_{f,t}Q_{f,t} + \Phi_{s,2}\Phi_{s,3}\delta D_{f,t-1}Q_{f,t-1} + \Phi^2_{s,2}\Phi_{s,3} \delta D_{f,t-2}Q_{f,t-2} -\Phi_{s,1}\delta D_{s,t-1}ET_{s,t}\}/Q_{s,t}$ $\delta D_{f,t}=\{\Phi_{f,1}(\delta D_{p-d,t}I_{f,t}+\delta D_{h,t}Q_{h-f,t})+\Phi_{f,2}\delta D_{f,t-1}Q_{f,t-1}+ \Phi_{f,3}(\Phi_{s,1}\delta D_{p-d,t}I_{s,t}+\Phi_{s,2}\Phi_{s,1}\delta D_{p-d,t-1}I_{s,t-1} +\Phi^2_{s,2}\Phi_{s,1}\delta D_{p-d,t-2}I_{s,t-2} +\Phi_{s,3}\delta D_{f,t}Q_{f,t} +\Phi_{s,2}\Phi_{s,3}\delta D_{f,t-1}Q_{f,t-1} + \Phi^2_{s,2}\Phi_{s,3}\delta D_{f,t-2}Q_{f,t-2}) - \Phi_{f,1} \delta D_{f,t-1}ET_{f,t}\}/(Q_{f,t} -\Phi_{f,3}\Phi_{s,3} Q_{s,t-2})$ | Full derivation process in Zhang et al., 2017 |
| $\Phi_{s,1} = 1/(K_s+fK_s/K_e)$ $\qquad \Phi_{f,1} = 1/(K_f+K_f/K_e+1)$ $\Phi_{s,2} = K_s/(K_s+ fK_s/K_e)$ $\qquad \Phi_{f,2} = K_f/(K_f+K_f/K_e+1)$ $\Phi_{s,3} = K_f/(K_sK_e+fK_s)$ $\qquad \Phi_{f,3} = fK_s/\{K_e(K_f+K_f/K_e+1)\}$ | |
| $Age_{s,t}=\{\Phi_{s,1}(Age_{p,t}I_{s,t}+Age_{h,t}Q_{h-s,t})+\Phi_{s,2}\Phi_{s,1}((Age_{h,t-1}+1)I_{s,t-1}+(Age_{h,t-1}+1)Q_{h-s,t-1})+\Phi^2_{s,2}\Phi_{s,1}((Age_{p,t-2}+2)I_{s,t-2}+(Age_{h,t-2}+2)Q_{h-s,t-2})+\Phi^2_{s,2}(Age_{s,t-2}+2)Q_{s,t-2}+ \Phi_{s,3}Age_{f,t}Q_{f,t} + \Phi_{s,2}\Phi_{s,3}(Age_{f,t-1}+1)Q_{f,t-1} + \Phi^2_{s,2}\Phi_{s,3}(Age_{f,t-2}+2)Q_{f,t-2} -\Phi_{s,1}(Age_{s,t-1}+1)ET_{s,t}\}/Q_{s,t}$ | |
| $Age_{f,t}=\{\Phi_{f,1}(Age_{p,t}I_{f,t}+Age_{h,t}Q_{h-f,t})+\Phi_{f,2}(Age_{f,t-1}+1)Q_{f,t-1}+\Phi_{f,3}(\Phi_{s,1}Age_{p,t}I_{s,t}+\Phi_{s,2}\Phi_{s,1}(Age_{p,t-1}+1)I_{s,t-1}+\Phi^2_{s,2}\Phi_{s,1}(Age_{p,t-2}+2)I_{s,t-2}+\Phi_{s,3}Age_{f,t}Q_{f,t}+ \Phi_{s,2}\Phi_{s,3}(Age_{f,t-1}+1)Q_{f,t-1} + \Phi^2_{s,2}\Phi_{s,3}(Age_{f,t-2}+2)Q_{f,t-2})- \Phi_{f,1}(Age_{f,t-1}+1)ET_{f,t}\}/(Q_{f,t}-\Phi_{f,3}\Phi_{s,3}Q_{s,t-2})$ | |


**Table A2 Description of the calibrated parameters**

| Coefficient | Units | Descriptions |
|---|---|---|
| $K_s$ | hour | The slow flow reservoir constant |
| $K_f$ | hour | The fast flow reservoir constant |
| $K_e$ | hour | Exchange constant between the two reservoirs |
| f | - | The ratio of porosity of the quick to slow flow reservoir |
| a | - | Precipitation recharge coefficient for slow flow reservoir |
| w | - | The hillslope unit constant |
| b | - | Recharge coefficient of Hillslope to slow flow reservoir |
| Is | - | Coefficient for evaporation fractionation |
| KK | - | Constant for calculation of rainfall recharging the active store in |
| pp | - | hillslope |
| con | - | Coefficient for exchange flow between active and passive stores in hillslope |
| fei | - | Weighting constant |