# Peer review of "Storage dynamics, hydrological connectivity and flux ages in a karst catchment: conceptual modelling using stable isotopes"

_Hydrology and Earth System Sciences, 2018_

## Referee Comment (RC1) · Anonymous Referee #1 · 6 May 2018

This paper presents some interesting simulations of a karst catchment in China. However, (at present) I cannot recommend publication, but after the following concerns are addressed

However, before I can recommend publication the following list of concerns need to be addressed.

Main comments

From reading this paper, it is unclear what the real novel contribution is. Surely interesting results are presented, but what do we really learn? I cannot derive this from the abstract, nor the conclusions. Please make this MUCH more explicit. The specific aims

tell you mostly "what" you do, instead of what you want to learn (and what is new about that). Only once I know what we aim to learn from this paper I can properly review the paper. Right now I mainly see a long list of results and statements. Sure I could comment on every detail of them, but that would not warrant a review which allows me to judge the scientific contribution of this paper well.

The writing of this paper needs significant improvement. In its current format, the paper contains very awkward and confusing use of the English language, which makes it at times hard to read and review. I suggest a native speaker takes a critical look at the whole paper. That makes more sense than that the reviewer does all this work for them. Nevertheless, I provide a long list of suggestions below, but addressing these will probably be not sufficient to tackle the language problems of this paper. Note that these problems with the writing do not only refer to grammar issues, but also to the plethora of statements, structure of reasoning, etc. that are unclear it the current format.

Detailed comments

Line 9: "unique" does not seems appropriate since other studies have similar or higher temporal resolution isotope and hydrometric data. For example,

Floury, P., Gaillardet, J., Gayer, E., Bouchez, J., Tallec, G., Ansart, P., Koch, F., Gorge, C., Blanchouin, A., and Roubaty, J.-L.: The potamochemical symphony: new progress in the high-frequency acquisition of stream chemical data, Hydrol. Earth Syst. Sci., 21, 6153-6165, https://doi.org/10.5194/hess-21-6153-2017, 2017.

von Freyberg, J., Studer, B., and Kirchner, J. W.: A lab in the field: high-frequency analysis of water quality and stable isotopes in stream water and precipitation, Hydrol. Earth Syst. Sci., 21, 1721-1739, https://doi.org/10.5194/hess-21-1721-2017, 2017.

Line 10: "flow-tracer model" is not really a clear term

Line 10: the model represents "the movement of water" using "two main landscape . . ..".

I suggest to add this, otherwise the sentence does not make much sense anymore.

Line 11: "cock-pit": I think you can remove the hyphen.

Line 12: "this inferred" is not logical. Something like "from these model results we inferred" would be much better

Line 13: or something like "had least water stored, whereas the slow reservoir has least water stored" (which makes the sentence more understandable, and it removes the redundant "intermediate" part.

Line 14: specify that you talk about mean ages OF WATER.

Line 14: "marked" seems unclear and redundant to me

Line 14-16: This statement is somewhat meaningless with its current explanation. "Connectivity can be defined in many ways" so I suggest that you describe what you physically found, rather than use an undefined buzzword. Actually, all the statements until sentence 18 are somewhat unclear. What do you mean by "reversible directionality"? I can guess, but please try to make the wording clearer to the reader.

Line 16-19: please revisit these sentences to make this an understandable abstract.

Line 32: "whole catchment" instead of "whole karst system" (the karst system may have a different scale).

Line 33: "However, semi-distributed lumped models need to have hydrogeological units adequately represented, in order to relate water flow in different landscape units and model parameters that have physically meaningful concepts." Is not logically connected to the previous statements. Where does the "however" come from?

Line 36: "Three main types of porosities – (a) micropores, (b) small fractures, and (c) large fractures and conduits – can be intuitively identified in karst systems." Do would it not help to start a new paragraph here?

Line 35: "can be intuitively identified" what do you mean here?

Line 42: (Rimmer and Hartmann, 2012; Hartmann et al, 2014; Zhang et al, 2017). Include and "e.g. since many more examples will exist).

Line 43-46: please rephrase "However, this kind of approach cannot disaggregate water storage and 45 flux dynamics within different landscape units, and may be inadequate for modelling when understanding known spatial differences in hydrogeological structure is important in terms of provisioning water supplies and understanding water quality issues (Fu et al, 2016; Zhang et al, 2013)"

Line 59: I think what Kirchner said is that these tracers help to 'highlight their differencs" rather than that they "resolve" anything really.

Line 71: "Hydrological connectivity, which has been simply defined as the transfer of water from one part of the landscape to another (McGuire and McDonnell, 2010; Golden et al, 2014; Soulsby et al., 2015)," this statement suggests that hydrologic connectivity is about the transport of water (e.g. velocity) rather than the "celerity effects" it is used for to describe. I think you need to be more accurate in its description.

Section 2.1. Did you take this information from other (peer reviewed) publications? If yes, please cite these.

Figure 1: please make it much more explicit in the caption what you display here.

Table 1: the range is a redundant variable.

Table 1: consider indicating how much of the time there is zero flow.

Table 1: why not provide a flow duration curve instead. That will we WAY more informative than what you currently present.

Line 159: CalculationS

Line 162: for each of the (not in each of the)

162-163: inconsistent with singular and plural. Check grammar.

Line 168: fix superscript "rainfall (m3 hour-1)"

Equations 8-11: I presume you talk about some mean age for the box, please specify this.

Equations 8-11 there equations are missing the "aging" term. (i.e. water gets older over time), please add this term and check if you calculations are correct...

Section 3.2 months spin up time may be sufficient spin up time for hydrometric fluxes, but will it be for modeling of ages?

Section 3.2: "First, different parameter combinations within the initial ranges in Table 3 were tested. And then, the parameter ranges were reduced according to the best models (KGE >0.3) for the second calibration. This resulted in a total of 10^5tested different parameter combinations. I do not understand how you arrive at the second 10^5.

Line 276: "rogue" ? what do you mean

Figure 10: these values cannot be correct since the areas under these curves do not add up to 1.

Line 420: cannot instead of can't.

Line 445-447 "Given the results on water storage dynamics and the relative contribution to the fast flow reservoir shown in Figures 7 and 8, it can be deduced that the storage change within each conceptual store is the main driver of hydrological connectivity between them." Is this not just how you defined that the catchments functions yourself? So what did we really learn in the end? (also remove the "s" in stores)

---

## Referee Comment (RC2) · Anonymous Referee #2 · 12 Jul 2018

In the submitted manuscript, Zhang et al present a modelling studies that deal with water dynamics, water isotope ratios and water ages in a small karst system in South West China. The authors use a lumped modelling approach that is able to simulate ïĄđ$^2$H ratios using mixing and partial mixing assumptions. Using the same approach, the authors also calculate water ages for the three main storages of the model, which are a hillslope storage, a fast karst groundwater reservoir and a slow karst groundwater reservoir. Using a Monte Carlo approach, the authors provide simulations discharge and ïĄđ$^2$H of their model including uncertainty ranges. Using regional sensitivity analysis, they show that 5 of their 12 model parameters are sensitive. In the following, the authors analyse the model internal dynamics to better understand landscape connectivity and age distributions in the system's subunits.

1. The study is well-written and concise. The model calibration and sensitivity analysis are detailed and well described However, there are some remaining concerns that the authors may account for before their manuscript can be considered for publication:

2. For the reader, who is not familiar with the author's preceding work, it is not clear how the model works. The schematic description in Fig 2 indicates that ET is taking place from the slow and fast karst groundwater storages, which would be quite unusual. To avoid misconception, please provide a complete model description in appendix (the table A1 is hardly understandable).

3. Some clarification on where the novelties of this work start is necessary. The authors inform the reader that in Zhang et al. (2017), the model was developed in previous work that used tracer data in addition to stream discharge to constrain the model structure, improve parameterization, and aid calibration. If this was done before, and the methods only describe how the isotope enabled model was parametrized and evaluated, what is the novelty of this particular study?

4. Figure 4 shows that only 5 of 12 parameters are sensitive, which is quite a low number. Usually, discharge contains enough information to identify 4-6 parameters (Jakeman & Hornberger, 1993). Adding of additional information like isotopes should increase this number, if the model structure is well-chosen. To check the contribution of discharge data and isotopes, could the authors show the parameter sensitivities using discharge or water isotopes only?

5. With a large fraction of the model parameters insensitive, how conclusive are the interpretations on the model internal dynamics that the authors use to explain connectivity and water age distribution in the system? In some of the figures, uncertainty ranges are provided and they are quite wide. In other figures (e.g., Fig. 5), only the mean is provided although the parameters controlling the observed processes ("w" in case of Fig. 5) are insensitive.

I am confident that the authors are able to address these moderate remarks and I am looking forward to reading a revised version of the manuscript. Some more technical and specific comments can be found in the attached pdf.

Please also note the supplement to this comment:
https://www.hydrol-earth-syst-sci-discuss.net/hess-2018-205/hess-2018-205-RC2-supplement.pdf

[Figure]

**Supplement:**

[revised manuscript text omitted]

---

## Author Comment (AC1) · 9 Aug 2018

This paper presents some interesting simulations of a karst catchment in China. However, (at present) I cannot recommend publication, but after the following concerns are addressed. However, before I can recommend publication the following list of concerns need to be addressed.

We sincerely thank the reviewer for his/her comments and suggestions that will help us improve our manuscript. We will thoroughly revise our manuscript taking into account all suggestions and comments from the reviewer. Our point-to-point responses are detailed below.

**Main comments**

From reading this paper, it is unclear what the real novel contribution is. Surely interesting results are presented, but what do we really learn? I cannot derive this from the abstract, nor the conclusions. Please make this MUCH more explicit. The specific aims tell you mostly "what" you do, instead of what you want to learn (and what is new about that). Only once I know what we aim to learn from this paper I can properly review the paper. Right now I mainly see a long list of results and statements. Sure I could comment on every detail of them, but that would not warrant a review which allows me to judge the scientific contribution of this paper well.

Reply: We will revise the manuscript to emphasize more clearly the novel contribution of the manuscript as follows:

(1) For cockpit terrain in the southwest China karst area, hillslope runoff processes are mostly routed into depression aquifers prior to contributing to streamflow ("hillslope- to- depression- to-stream"). Identifying and quantifying the dynamics of water storage, hydrological connectivity between different stores and the associated ages of water fluxes is very important to understand how the unique landscape characteristics of karst affect flow transmission. This has applied significance for understanding water resource availability and flood hazard management.

(2) Consequently, we developed a tracer-aided runoff model that disaggregates the cockpit karst terrain into the two dominant landscape units of hillslopes and depressions (further sub-dividing the depression into fast and slow reservoirs) extending an earlier model developed by the authors of the dual flow reservoirs for flow and solute ($Ca$ and $Mg$) concentrations at the catchment scale. This tracer-aided model conceptualizes hydrological functions more comprehensively by estimating storage-flux dynamics and water ages in each unit. Such tracer-aided models enhance our understanding of the hydrological connectivity between different landscape units and the mixing processes between various flow sources.

(3) Since the tracer-aided model increases model parametersisation for the tracer modules, we evaluated the uncertainty of the modelled results, including not only those of flow and stable isotopic values, but also water storage and flux ages at various landscape units. In particular,, we found that the tracer-aided models can be used to characterize the uncertainty of modelled results at any units in the catchment.

The writing of this paper needs significant improvement. In its current format, the paper contains very awkward and confusing use of the English language, which makes it at times hard to read and review. I suggest a native speaker takes a critical look at the whole paper. That makes more sense than that

the reviewer does all this work for them. Nevertheless, I provide a long list of suggestions below, but addressing these will probably be not sufficient to tackle the language problems of this paper. Note that these problems with the writing do not only refer to grammar issues, but also to the plethora of statements, structure of reasoning, etc. that are unclear it the current format.

Reply: We appreciate the referee's suggestions, and the whole paper will be thoroughly revised to improve the clarity and grammar.

**Detailed comments**

Line 9: "unique" does not seems appropriate since other studies have similar or higher temporal resolution isotope and hydrometric data. For example,

Floury, P., Gaillardet, J., Gayer, E., Bouchez, J., Tallec, G., Ansart, P., Koch, F., Gorge, C., Blanchouin, A., and Roubaty, J.-L.: The potamochemical symphony: new progress in the high-frequency acquisition of stream chemical data, Hydrol. Earth Syst. Sci., 21, 6153-6165, https://doi.org/10.5194/hess-21-6153-2017, 2017.

von Freyberg, J., Studer, B., and Kirchner, J. W.: A lab in the field: high-frequency analysis of water quality and stable isotopes in stream water and precipitation, Hydrol.Earth Syst. Sci., 21, 1721-1739, https://doi.org/10.5194/hess-21-1721-2017, 2017.

Reply: We will re-state this. Whilst we recognize that others have such higher temporal resolution steam data, we wished to emphasize that our high resolution, extended isotope and hydrometric observations concurrently collected in hillslopes, depressions and streams of complex karst catchments are scarce.

Line 10: "flow-tracer model" is not really a clear term
Reply: This will be replaced by "tracer-aided model".

Line 10: the model represents "the movement of water" using "two main landscape ….".I suggest to add this, otherwise the sentence does not make much sense anymore.
Reply: We will revised this.

Line 11: "cock-pit": I think you can remove the hyphen.
Reply: We will remove the hyphen.

Line 12: "this inferred" is not logical. Something like "from these model results we inferred" would be much better.
Reply: We will revise this.

Line 13: or something like "had least water stored, whereas the slow reservoir has least water stored" (which makes the sentence more understandable, and it removes the redundant "intermediate" part.
Reply: We will revise this.

Line 14: specify that you talk about mean ages OF WATER.
Reply: We will revise this to. "The estimated mean ages of the hillslope unit, fast and slow flow reservoirs during the study period ……"

Line 14: "marked" seems unclear and redundant to me
Reply: The "marked" will be replaced by "highly"

Line 14-16: This statement is somewhat meaningless with its current explanation. "Connectivity can be defined in many ways" so I suggest that you describe what you physically found, rather than use an undefined buzzword. Actually, all the statements until sentence 18 are somewhat unclear. What do you mean by "reversible directionality"? I can guess, but please try to make the wording clearer to the reader.
Reply: We will revise the sentences from lines 14 to 19. We will clarify that the connectivity in our study refers to fluxes from "hillslope- to- depression (fast and slow reservoirs)- to- outlet stream".

Line 16-19: please revisit these sentences to make this an understandable abstract.
Reply: We will revise the sentences from line 16 to 19.

Line 32: "whole catchment" instead of "whole karst system" (the karst system may have a different scale).
Reply: We will revise this.

Line 33: "However, semi-distributed lumped models need to have hydrogeological units adequately represented, in order to relate water flow in different landscape units and model parameters that have physically meaningful concepts." Is not logically connected to the previous statements. Where does the "however" come from?
Reply: We will revise this paragraph. In the revised manuscript, we will first describe the lumped models and then the semi-distributed models.

Line 36: "Three main types of porosities – (a) micropores, (b) small fractures, and (c) large fractures and conduits – can be intuitively identified in karst systems." Do would it not help to start a new paragraph here?
Reply: Yes, we will revise this and define terms more precisely. In karst aquifers, the solutional conduits connect with intergranular pores and fractures (often termed as matrix porosity), showing dual or even triple porosity zones (Worthington, Jeannin, Alexander, Davies, & Schindel, 2017). Thus, karst aquifers are often conceptualized as dual porosity systems as residence times in the matrix are several orders of magnitude longer than those in the conduits (Goldscheider & Drew, 2007).

Line 37: "can be intuitively identified" what do you mean here?
Reply: We will delete the sentence in line 37 and replace with the statement above.

Line 42: (Rimmer and Hartmann, 2012; Hartmann et al, 2014; Zhang et al, 2017). Include and "e.g. since many more examples will exist).
Reply: We will revise this.

Line 43-46: please rephrase "However, this kind of approach cannot disaggregate water storage and

flux dynamics within different landscape units, and may be inadequate for modelling when understanding known spatial differences in hydrogeological structure is important in terms of provisioning water supplies and understanding water quality issues (Fu et al, 2016; Zhang et al, 2013)"
Reply: We will revise this.

Line 59: I think what Kirchner said is that these tracers help to 'highlight their differences" rather than that they "resolve" anything really.
Reply: We will revise this and replace the literature by the more precise descriptions from Birkel et al, 2015.

Birkel, C., Soulsby, C. and Tetzlaff, D.: Conceptual modelling to assess how the interplay of hydrological connectivity, catchment storage and tracer dynamics controls nonstationary water age estimates, Hydrol. Process., 29(13), 2956–2969, doi:10.1002/hyp.10414, 2015b.

Line 71: "Hydrological connectivity, which has been simply defined as the transfer of water from one part of the landscape to another (McGuire and McDonnell, 2010; Golden et al, 2014; Soulsby et al., 2015)," this statement suggests that hydrologic connectivity is about the transport of water (e.g. velocity) rather than the "celerity effects" it is used for to describe. I think you need to be more accurate in its description.
Reply: We will delete the sentence which gave the different descriptions of hydrologic connectivity than what we used.

Section 2.1. Did you take this information from other (peer reviewed) publications? If yes, please cite these.
Reply: We will add the relevant publications.

Figure 1: please make it much more explicit in the caption what you display here.
Reply: We will add the relevant descriptions.

Table 1: the range is a redundant variable.
Reply: We will revise this.

Table 1: consider indicating how much of the time there is zero flow.
Reply: We will add this item. It occupies only a short period of time in our observation period.

**Table 1 Statistical summary of flow discharge for hillslope spring (HS) and catchment outlet (m3/s)**

| Obs | Min | Max | Mean | Cv | Time with zero flow (h) |
|---|---|---|---|---|---|
| Outlet | 0 | 0.15 | $4.7 \times 10^{-3}$ | 2.83 | 328 |
| HS | 0 | $1.4 \times 10^{-3}$ | $8.5 \times 10^{-5}$ | 1.73 | 713 |

Table 1: why not provide a flow duration curve instead. That will we WAY more informative than what you currently present.
Reply: The flow duration curves will be added.

Line 159: CalculationS
Reply: We will revise this.

Line 162: for each of the (not in each of the)
Reply: We will revise this.

Line 162-163: inconsistent with singular and plural. Check grammar.
Reply: We will revise this sentence and make corrections in the whole manuscript.

Line 168: fix superscript "rainfall (m3 hour-1)" Equations 8-11: I presume you talk about some mean age for the box, please specify this. Equations 8-11 there equations are missing the "aging" term. (i.e. water gets older over time), please add this term and check if you calculations are correct: : :
Reply: We will revise this, and give a more complete description in the appendix.
We have considered the "aging" item. In the model procedure, each age item at the time t includes the age at the previous time step t-1. So, the results listed in this paper include the "aging effect" (this will be clarified in the appendix in an attached file about the model descriptions).

Section 3.2 months spin up time may be sufficient spin up time for hydrometric fluxes, but will it be for modeling of ages?
Reply: The spin up time for the modeling of ages is sufficient given the young water dominance. Our two step calibration procedures show: as the mean value of the modelled water ages (meeting the target of KGE>0.3) in the first calibration were used as the initial water ages for the second calibration, the calibrated water ages for each conceptual store well matches the measured isotope values (the target of KGE increases to be higher than 0.5). It means that the selected initial period for "warm up" modeling of ages is reasonable.

Section 3.2: "First, different parameter combinations within the initial ranges in Table 3 were tested. And then, the parameter ranges were reduced according to the best models (KGE >0.3) for the second calibration. This resulted in a total of 10ˆ5 tested different parameter combinations. I do not understand how you arrive at the second 10ˆ5.
Reply: We will revise the descriptions and add the initial parameters from the first calibration. A total of 10ˆ5 different parameter combinations was given for the estimation of uncertainty of the modeled results from the random generation of the possible parameter combinations. The number of 10ˆ5 different parameter combinations is believed to be sufficient according to uncertainty analysis in the literatures (Soulsby et al., 2015; Xie et al., 2017). In the first and second calibrations, the number of parameter combinations were set to 10ˆ5, but the range of the initial parameters are different (Initial range 1 and 2 for the 1st and 2nd calibration in revised Table 3).
After the first calibration when KGE >0.3, the range of each parameter was reduced. Then, the narrowed ranges (initial range 2 in Table 3) were used as the initial ranges for the second calibration.

Table 3

| For Flow | $K_s$ (hour$^{-1}$) | $K_f$ (hour$^{-1}$) | $K_e$ (hour$^{-1}$) | $f$ | $a$ | $W$ | $b$ |
|---|---|---|---|---|---|---|---|
| Initial range 1 | 40-168 | 1-72 | 800-2200 | 0.005-0.025 | 0-1 | 0-0.015 | 0-1 |
| Initial range 2 | 40-150 | 1-40 | 800-2200 | 0.008-0.025 | 0.47-1 | 0-0.015 | 0.48-1 |
| Mean | 92 | 11 | 1549 | 0.015 | 0.68 | 0.005 | 0.54 |
| Range | 48-120 | 5-18 | 1000-2000 | 0.01-0.02 | 0.51-0.9 | 0.003-0.01 | 0.5-0.62 |

| For Isotope | $Is$ | $KK$ ($\times 10^4$) | $pp$ | $con$ | $fei$ | Index | Mean(range) |
|---|---|---|---|---|---|---|---|
| Initial range 1 | 0-1 | 0.8-1.6 | 0-1 | 0-1 | 0-1 | $KGE_d$ | 0.85 (0.81-0.87) |
| Initial range 2 | 0-0.8 | 0.8-1.6 | 0-1 | 0-1 | 0.5-1 | $KGE_i$ | 0.56 (0.52-0.59) |
| Mean | 0.24 | 1.26 | 0.49 | 0.56 | 0.82 | $KGE$ | 0.7 (0.72-0.66) |
| Range | 0.002-0.6 | 1-1.5 | 0.02-0.95 | 0.04-0.97 | 0.71-0.93 | | |

Line 276: "rogue" ?   what do you mean
Reply: It refers to some samples that are unusually high during the study period (in Fig.3). These samples could be affected by the paddy water in the manuscript description.

Figure 10: these values cannot be correct since the areas under these curves do not add up to 1.
Reply: The figure gave the probability density functions (PDFs) of the flux ages from the three units. The sum of these values for the three units, e.g. the total areas covered by the three curves with different colors (conceptual stores) equals to 1.

Line 420: cannot instead of can't.
Reply: We will revise this.

Line 445-447 "Given the results on water storage dynamics and the relative contribution to the fast flow reservoir shown in Figures 7 and 8, it can be deduced that the storage change within each conceptual store is the main driver of hydrological connectivity between them." Is this not just how you defined that the catchments functions yourself? So what did we really learn in the end? (also remove the "s" in stores)
Reply: We will revise the conclusions. This description will be changed to: for the "hillslope-depression-outlet" connectivity, the weak/strong connectivity between hillslope and depression was inferred by only a small/large percentage (16.8% and 57.5%) of the outlet fluxes from the hillslope unit contribution during the dry/wet period (Fig 8). The seasonal decline and increase of the fast flow storage were primarily maintained by the slow flow while the short-term variability of the fast flow storage responds to the hillslope flow (Fig 7) as well as rainfall recharge (Fig 8).

---

## Author Comment (AC2) · 9 Aug 2018

**Main comments**

1. The study is well-written and concise. The model calibration and sensitivity analysis are detailed and well described However, there are some remaining concerns that the authors may account for before their manuscript can be considered for publication:
We sincerely thank the reviewer for carefully reviewing our manuscript and for the thoughtful, constructive feedback.

2. For the reader, who is not familiar with the author's preceding work, it is not clear how the model works. The schematic description in Fig 2 indicates that ET is taking place from the slow and fast karst groundwater storages, which would be quite unusual. To avoid misconception, please provide a complete model description in appendix (the table A1 is hardly understandable).

Reply: In this model, the karst critical zone in the hillslope was conceptualized as one reservoir, but the water stored in the reservoir was further sub-divided into upper active and lower passive storage zones (Fig 2) for the simulation of isotope ratios and estimation of water ages. This division follows our previous measurements of the vertical distribution of the rock fracturs/conduits along hillslopes where the large rock fracturs/conduits decrease exponentially in the vertical direction (Zhang et al., 2011).

The karst critical zone in the depression was conceptualized as two connected reservoirs, fast and slow flow, representing the solutional conduits in karst aquifers connecting with intergranular pores and fractures (often termed as matrix porosity).

The evapotranspiration could occur from the rich conduit/fracture areas by extended plant roots in the deep aquifer (Rong et al., 2011). Therefore, evapotranspiration is sourced from both the fast and slow reservoirs in the model.

We will revise the model descriptions in an appendix for clearer explanation of the module functions and meanings in the revised manuscript.

3. Some clarification on where the novelties of this work start is necessary. The authors inform the reader that in Zhang et al. (2017), the model was developed in previous work that used tracer data in addition to stream discharge to constrain the model structure, improve parameterization, and aid calibration. If this was done before, and the methods only describe how the isotope enabled model was parametrized and evaluated, what is the novelty of this particular study?

Reply: The model in the preceding work (Zhang et al., 2017), conceptualized the flow and the geochemical solute (Ca+Mg) routings using conceptualization of the dual flow system at the catchment scale. So, the original model had no basis for disaggregating the hydrological connectivity between different landscape units (e.g. "hillslope- to- depression- to- stream" in the study catchment). The hillslope-depression is a typical landform with variable hydrological connectivity in the karst catchments in southwest of China (Figure r1, Chen et

al., 2018). Here, we improved our previous model structure by conceptualizing the hillslope and depression units (the improved part is in the red dotted box in Figure r2), and then use the hourly discharge and isotope values to calibrate the model. In addition, the new model has the parameters to represent passive storage inferred by isotope damping and the function of estimating the water ages from various landscape units in the catchment.

Although the tracer-aided model enhanced our understanding of the hydrological connectivity between different landscape units and the mixing processes, it increased the model parameters in the tracer modules. Therefore, we also evaluated the uncertainty of the simulation results including flow discharges, isotopic values, storages and ages at the different landscape units in this study.

[Figure]

Figure r1 Sketch map of karst hydrological processes (Chen et al, 2018)

[Figure]

Figure r2 Structure of the improved model, and the improved part is in the red dotted box

4. Figure 4 shows that only 5 of 12 parameters are sensitive, which is quite a low number. Usually, discharge contains enough information to identify 4-6 parameters (Jakeman & Hornberger, 1993). Adding of additional information like isotopes should increase this number, if the model structure is well-chosen. To check the contribution of discharge data and isotopes, could the authors show the parameter sensitivities using discharge or water isotopes only?

Reply: The trace-aided model includes 12 parameters, seven for flow routing (*Ks, Kf, Ke, f, a, w,* and *b)* and five for isotope ratios and water ages *(Is, KK, pp, con* and *fei)*. So, the overall model increased by five parameters in the isotopic module.

We analyzed the parameter sensitivities using either the outlet discharge and/or water isotopes. Targeting the discharge, six parameters (except w) among the seven parameters in the flow routing module are sensitive and the parameters in the isotopic module are all insensitive (Fig r3 (a)). Targeting only isotopic values and both flow discharge and isotopic composition, the sensitive parameters are same, including *Kf, a,* and *b* in the flow routing module, and *Is* and *fei* in the tracer module) (Fig r3 (b) and (c)). Using both flow discharge and isotopic composition as the target, these parameters were more sensitive than those using only isotopic values (see the wide ranges of the cumulative distributions in Fig r3 (c)).

Interestingly, increasing the two sensitive parameters in the isotopic module (the coefficient for evaporation fractionation *Is* and the weighted isotope composition of rainfall input by the parameter *fei*) results in three parameters in the flow module becoming insensitive (slow reservoir constant (*Kf*), the exchange constant between the two reservoirs *Ke* and the ratio of porosity of the quick to slow flow reservoir *f*).

This can be explained as follows: the former two sensitive parameters in the isotopic module emphasize atmospheric effects on the outlet flow (being "old/new"). Larger *Is* indicates more evaporative effect on the stored water, leading to the stored and released water being older, particularly during the dry period. Larger *fei* indicates newer rainfall recharge (more negative isotopic values) into aquifer, leading to the stored and released water being newer during rainfall period. Alternatively, the latter three parameters in the flow module emphasize effects of fast (newer) and slow (older) flows in aquifer on the outlet flow (being "old/new"). More water release from the slow reservoir (larger *Kf*) and greater release of the slow reservoir into the fast reservoir (larger *Ke*) could lead to the released water being older in the dry season; a high proportion of the fast flow storage (larger *f*) and a greater exchange between the fast reservoir and the slow reservoir (larger *Ke*) could lead to the released water being newer in the wet season. Consequently, there is equifinality for these parameters in the trace-aided model, which can be overcome only when we have additional data to constrain some of the parameters, e.g. knowing the evaporative effect on water *Is* and the weighted isotope composition of rainfall input by the parameter *fei*.

We will add the above reasoning in our discussion of the revised manuscript.

[Figure]

(a) Sensitive parameters include Ks, Kf, Ke, f, a, and b

(b) Sensitive parameters include Kf, a, b, Is and fei

[Figure]

(c) Sensitive parameters include Kf, a, b, Is and fei

Figure r3 Sensitivity of 10 model parameters using (a) flow, (b) isotope composition and (c) combined simulation of flow and isotopic composition. (The parameters inside the gray dotted box are for flow routing, and the outside parameters are for isotope routing.)

5. With a large fraction of the model parameters insensitive, how conclusive are the interpretations on the model internal dynamics that the authors use to explain connectivity and water age distribution in the system? In some of the figures, uncertainty ranges are provided and they are quite wide. In other figures (e.g., Fig. 5), only the mean is provided although the parameters controlling the observed processes ("w" in case of Fig. 5) are insensitive.

Reply: In the revised manuscript, we will describe the uncertainty of the modeled results for the various landscape units in the catchment. We reached the following conclusions:

The outlet hydrometric and isotope observations (consisting of mostly young and fast flows) were used as the calibration targets in this study. The outlet simulations had the least uncertainty, while uncertainty in the hillslope and depression units were highly related to their hydrological connectivity with the outlet. The simulated fast flows in the hillslope and depression units had lower uncertainty than the simulated slow flows in the depression since the two former units are highly connect with the outlet.

Although some parameters (e.g. w controlling hillslope flow dynamics) are insensitive, uncertainty bands of the hillslope flow (Fig r4) are narrow and the model captures quite well the hillslope seasonality and event-based dynamics through targeting the best matching of outlet discharges and isotopic values. This indicates that the hillslope dynamics are closely

linked to the outlet dynamic patterns (with strong connectivity between them), which is consistent with the ranges of δD and δ¹⁸O values at the hillslope spring being close to the ranges at the outlet discharge in Table 2.

For some of the other insensitive parameters (e.g. *Ke* that determines the exchange amount between the fast and slow flow reservoirs), uncertainty of the simulated exchange flux is much higher than that of the water fluxes from direct rainfall recharge and hillslope fluxes (see Figure r5), indicating weaker connectivity of the slow flow with the outlet flow.

In the revised manuscript, we will also present analysis of uncertainties of the modelled storages and ages (using the new capability of the tracer-aided model) at the various units. This shows that: the uncertainty increases with ages in comparison of the three water fluxes (Fig 9 in the original version of the manuscript), i.e. the narrowest uncertainty bands for the youngest hillslope flow and the widest for the oldest slow flow. Seasonal change in uncertainty also increases with age for the younger hillslope flow and the fast flow in depression. However, for the slow flow in the depression, change in the uncertainty decreases with ages, e.g. the bands tend to be wide for the younger water during rainfall season in Fig 9. The greater uncertainty as the slow flow becomes younger also reflects that the uncertainty is likely explained by the insensitive parameter of *Ke* (reducing effect of the frequent exchanges between fast and slow reservoirs on the outlet flow).

[Figure]

Figure r4 Observed discharge at hillslope spring against the simulated discharge of hillslope unit (values are normalized)

**Detailed comments**

Line 66: There are a few studies on water storage, flux and age dynamics using tracers in karst environments.
Reply: we will revise this expression.

Line 132: If this was done before, what is the novelty of this study?
Reply: See response to Q3 in the main comments above.

Line 145: Is there are distinction between soil/epikarst and groundwater? What controls matrix-conduit exchange?
Reply: We will revise the descriptions. In this model, we conceptualized the groundwater aquifer in the depression by a dual flow system (involving fast and slow flow reservoirs), and the groundwater aquifer in the hillslope by an upper active storage (mostly from epikarst) mixing with a lower passive storage since the rock fractures/conducts reduce with depth from the ground surface in the hillslope profile according to our previous investigations.

The exchange between matrix and conduit is controlled by the water storage (relate to water level) and the exchange constant between the two reservoirs ($Ke$) in each reservoir.

Line 222: This is not correct - please remove
Reply: We will revise this.

Line 260: typo
Reply: We will revise this.

Line 296: only 5 of 12 parameters are sensitive, which is quite a low number. Usually, discharge contains enough information to identify 4-6 parameters (Jakeman & Hornberger, 1993). Adding of additional information like isotopes should increase this number, if the model structure is well-chosen. To check this, could the authors show the parameter sensitivities using discharge only?
Reply: see the reply to Q4 in the main comments above.

Line 299: In the text, a rejection limit of 0.3 is mentioned. Please clarify
Reply: Two step calibrations were carried out in this study. First, $10^5$ different parameter combinations were selected with the broad ranges of initial parameter values. And then, we obtained the narrower ranges of the parameters according to the best modelled results (meeting the KGE >0.3 criteria). For the second calibration, the narrowed ranges of the parameters were used as the initial ranges of the parameter to search the next best modelled results ( KEG >0.5). We will revise the descriptions in the new manuscript accordingly.

Line 308: The parameter w is completely insensitive meaning that this storage's dynamics are not well identifiable, right? Please provide all behaviorals instead of the mean to show the precision of simulation of its discharge
Reply: See the reply to Q5 in main comments above.

Line 324: how much can you conclude from such wide uncertainty ranges?
Reply: the greater uncertainty of the modelled isotopic values in the depression arose from the insensitive parameters of $Ke$ and $Ks$ that affect the slow flow discharge and its exchange with the fast flow when the outlet hydrometric and isotope observations (consisting of mostly young and fast flows) used as metrics for the objective function for model calibration.

Here, the modelled isotope composition in the depression (see Figure 6b) refers to the release of water from the slow flow reservoir, representing a relatively constant source. The uncertainty bands can cover the limited variability of the measured values of δD at W1 and W5 (blue and yellow points in Figure 6b) where the aquifer has much lower permeability (W5) and is confined (W1) (cf the geophysical survey reported by Chen et al, 2018). This means that our tracer-aided model capture the slow flow dynamics in the depression even though the uncertainty is large.

The highly negative values of δD at W3 and W4 (red and black points in Figure 6b) are mostly below the uncertainty bands. This means that the stored water at W3 and W4 was younger than water from the slow flow reservoir, which is consistent with recent geophysical evidence (see Chen et al, 2018). Since W3 and W4 are located at high permeability areas, water at W3 and W4 was contributed mostly by fast flows (mixing with the young water), particularly during rainfall events (e.g. 9/7, and 20/7 in Fig 6b). So the high negative values of δD at W3 and W4 below the uncertainty bands were reasonable. We will revise the descriptions accordingly in the new manuscript.

Line 368: KE is also quite insensitive. Can you also show the entire 500 ensamble (or confidence limits)?

Reply: The simulations from the entire 500 ensemble are shown in Fig r5. Since the parameter *Ke* that determines the exchange amount between the fast and slow flow reservoirs is insensitive, the simulated exchange flux is highly uncertainty, though much smaller, compared to the water fluxes from the rainfall recharge and hillslope flow.

[Figure]

Figure r5 Source contributions to the underground stream flow (fast reservoir) at the

catchment outlet. The red dots above and under the dotted line represent transient reverse water fluxes from the slow reservoir to fast reservoir and fast reservoir to slow reservoir, respectively.

Line 387: Please double-check this with literature values. Fast flow components in karst systems provide water with ages mostly between days or weeks (including temporal storage in the epikarst). Mostly, the ages found here are too large, even in the wet period.

Reply: We believe that the estimated ages are reasonable. Most models do not include mixing processes with stored water so tend to under-estimate water ages.

Here we listed $\delta$D values at the sampling points in this catchment for the two largest rainfall events in 2017 (the details refer to Chen et al., 2018, https://doi.org/10.1002/hyp.13232).

| Date | rainfall amount | rain water | outlet water | hillslope spring |
|------|-----------------|------------|--------------|------------------|
| 12/6 | 86.6 mm | -85 | -48 ~ -70 | -62~-67 |
| 9/7 | 83.4mm | -80 | -62~ -73 | -59~ -70 |

It shows that $\delta$D values at outlet and hillslope spring are much less negative than rainwater. So there was strong mixing of the "new" rainwater with "old" stored water during and after the rainfall although the response of discharge to rainfall is fast.
Also, our estimated ages in the manuscript refer to the mean of the ages over a long period of time. For short-term (event based) responses to the rainfall, the ages of water from hillslope flow and fast reservoirs can be shortest as 4 and 2 days, respectively. There were 8 and 23 events for the fast flow with the ages of water less than 5 and 10 days, respectively (see the lowest values in Fig 9). So, the results are not inconsistent with previous work, rather capture the time-variance of water ages. We will add these explanations in the revised manuscript.

Line 398: See comment above
Reply: The same response as for Line 387.

Line 409: Some recent example how this can be done with water quality data in karst:
Hartmann, A., Barberá, J. A., & Andreo, B. (2017). On the value of water quality data and informative flow states in karst modelling. Hydrology and Earth System Sciences, 21, 5971–5985.
https://doi.org/10.5194/hess-2017-230
Reply: We will add this relevant literature.

Line 441: Large fractions of the fast reservoir have ages larger than several months, which appears a bit slow. (see also comments above)
Reply: See response to Line 387.

---

## Author Comment (AC3) · 9 Aug 2018

The comment was uploaded in the form of a supplement:
https://www.hydrol-earth-syst-sci-discuss.net/hess-2018-205/hess-2018-205-AC3-supplement.pdf

---

## Author Response (AR1)

This paper presents some interesting simulations of a karst catchment in China. However, (at present) I cannot recommend publication, but after the following concerns are addressed. However, before I can recommend publication the following list of concerns need to be addressed.

We sincerely thank the reviewer for his/her comments and suggestions that significantly improved our manuscript. We have thoroughly revised our manuscript taking into account all suggestions and comments from the reviewer. Our point-to-point responses are detailed below.

**Main comments**

From reading this paper, it is unclear what the real novel contribution is. Surely interesting results are presented, but what do we really learn? I cannot derive this from the abstract, nor the conclusions. Please make this MUCH more explicit. The specific aims tell you mostly "what" you do, instead of what you want to learn (and what is new about that). Only once I know what we aim to learn from this paper I can properly review the paper. Right now I mainly see a long list of results and statements. Sure I could comment on every detail of them, but that would not warrant a review which allows me to judge the scientific contribution of this paper well.

Reply: We have revised the manuscript to emphasize more clearly the novel contribution of the manuscript as follows:

(1) For cockpit terrain in the southwest China karst area, hillslope runoff processes are mostly routed into depression aquifers prior to contributing to streamflow ("hillslope- to- depression- to-stream"). Identifying and quantifying the dynamics of water storage, hydrological connectivity between different stores and the associated ages of water fluxes is very important to understand how the unique landscape characteristics of karst affect flow transmission. This has applied significance for understanding water resource availability and flood hazard management.

(2) Consequently, we developed a tracer-aided runoff model that disaggregates the cockpit karst terrain into the two dominant landscape units of hillslopes and depressions (further sub-dividing the depression into fast and slow reservoirs) extending an earlier model developed by the authors of the dual flow reservoirs for flow and solute (*Ca* and *Mg*) concentrations at the catchment scale. This tracer-aided model conceptualizes hydrological functions more comprehensively by estimating storage-flux dynamics and water ages in each unit. Such tracer-aided models enhance our understanding of the hydrological connectivity between different landscape units and the mixing processes between various flow sources.

(3) Since the tracer-aided model increases model parametersisation for the tracer modules, we evaluated the uncertainty of the modelled results, including not only those of flow and stable isotopic values, but also water storage and flux ages at various landscape units. In particular, we found that the tracer-aided models can be used to characterize the uncertainty of modelled results at any units in the catchment.

The writing of this paper needs significant improvement. In its current format, the paper contains very awkward and confusing use of the English language, which makes it at times hard to read and review. I suggest a native speaker takes a critical look at the whole paper. That makes more sense than that

the reviewer does all this work for them. Nevertheless, I provide a long list of suggestions below, but addressing these will probably be not sufficient to tackle the language problems of this paper. Note that these problems with the writing do not only refer to grammar issues, but also to the plethora of statements, structure of reasoning, etc. that are unclear it the current format.

Reply: We appreciate the referee's suggestions, and the whole paper has been thoroughly revised to improve the clarity and grammar.

**Detailed comments**

Line 9: "unique" does not seems appropriate since other studies have similar or higher temporal resolution isotope and hydrometric data. For example,

Floury, P., Gaillardet, J., Gayer, E., Bouchez, J., Tallec, G., Ansart, P., Koch, F., Gorge, C., Blanchouin, A., and Roubaty, J.-L.: The potamochemical symphony: new progress in the high-frequency acquisition of stream chemical data, Hydrol. Earth Syst. Sci., 21, 6153-6165, https://doi.org/10.5194/hess-21-6153-2017, 2017.

von Freyberg, J., Studer, B., and Kirchner, J. W.: A lab in the field: high-frequency analysis of water quality and stable isotopes in stream water and precipitation, Hydrol.Earth Syst. Sci., 21, 1721-1739, https://doi.org/10.5194/hess-21-1721-2017, 2017.

Reply: We have re-stated this. Whilst we recognize that others have such higher temporal resolution steam data, we wished to emphasize that our high resolution, extended isotope and hydrometric observations concurrently collected in hillslopes, depressions and streams of complex karst catchments are scarce.

Line 10: "flow-tracer model" is not really a clear term

Reply: This has been replaced by "tracer-aided model".

Line 10: the model represents "the movement of water" using "two main landscape ….".I suggest to add this, otherwise the sentence does not make much sense anymore.

Reply: We have revised this.

Line 11: "cock-pit": I think you can remove the hyphen.

Reply: We have removed the hyphen.

Line 12: "this inferred" is not logical. Something like "from these model results we inferred" would be much better.

Reply: We have revised this.

Line 13: or something like "had least water stored, whereas the slow reservoir has least water stored" (which makes the sentence more understandable, and it removes the redundant "intermediate" part.

Reply: We have revised this.

Line 14: specify that you talk about mean ages OF WATER.

Reply: We have revised this to "The estimated mean ages of the hillslope unit, fast and slow flow reservoirs during the study period ……"

Line 14: "marked" seems unclear and redundant to me
Reply: The "marked" has been replaced by "distinct"

Line 14-16: This statement is somewhat meaningless with its current explanation. "Connectivity can be defined in many ways" so I suggest that you describe what you physically found, rather than use an undefined buzzword. Actually, all the statements until sentence 18 are somewhat unclear. What do you mean by "reversible directionality"? I can guess, but please try to make the wording clearer to the reader.
Reply: We have revised the sentences from lines 14 to 19. We have clarified what we physically found.

Line 16-19: please revisit these sentences to make this an understandable abstract.
Reply: We have revised the sentences from line 16 to 19.

Line 32: "whole catchment" instead of "whole karst system" (the karst system may have a different scale).
Reply: We have revised this.

Line 33: "However, semi-distributed lumped models need to have hydrogeological units adequately represented, in order to relate water flow in different landscape units and model parameters that have physically meaningful concepts." Is not logically connected to the previous statements. Where does the "however" come from?
Reply: We have revised this paragraph. In the revised manuscript, we have first described the lumped models and then the semi-distributed models. (see lines 48-65 in the revised manuscript)

Line 36: "Three main types of porosities – (a) micropores, (b) small fractures, and (c) large fractures and conduits – can be intuitively identified in karst systems." Do would it not help to start a new paragraph here?
Reply: We have revised this and defined terms more precisely. "In karst aquifers, the solutional conduits connect with intergranular pores and small fractures (often termed as matrix porosity), showing dual or even triple porosity zones (Worthington, Jeannin et al., 2017). Thus, karst aquifers are often conceptualized as dual porosity systems as residence times in the matrix are often several orders of magnitude longer than those in the conduits (Goldscheider & Drew, 2007)." (see lines 52-58 in the revised manuscript)

Line 37: "can be intuitively identified" what do you mean here?
Reply: We have deleted the sentence in line 37 and replaced with the statement above.

Line 42: (Rimmer and Hartmann, 2012; Hartmann et al, 2014; Zhang et al, 2017). Include and "e.g. since many more examples will exist).
Reply: We have revised this.

Line 43-46: please rephrase "However, this kind of approach cannot disaggregate water storage and

flux dynamics within different landscape units, and may be inadequate for modelling when understanding known spatial differences in hydrogeological structure is important in terms of provisioning water supplies and understanding water quality issues (Fu et al, 2016; Zhang et al, 2013)"

Reply: We have revised this.

Line 59: I think what Kirchner said is that these tracers help to 'highlight their differences" rather than that they "resolve" anything really.

Reply: We have revised this to "…… have helped to resolve this celerity-velocity dichotomy known as the "old water paradox".

Line 71: "Hydrological connectivity, which has been simply defined as the transfer of water from one part of the landscape to another (McGuire and McDonnell, 2010; Golden et al, 2014; Soulsby et al., 2015)," this statement suggests that hydrologic connectivity is about the transport of water (e.g. velocity) rather than the "celerity effects" it is used for to describe. I think you need to be more accurate in its description.

Reply: We have deleted the sentence which gave the different descriptions of hydrologic connectivity than what we used.

Section 2.1. Did you take this information from other (peer reviewed) publications? If yes, please cite these.

Reply: We have added the relevant publications.

Figure 1: please make it much more explicit in the caption what you display here.

Reply: We have added the relevant descriptions.

Table 1: the range is a redundant variable.

Reply: This table has been replaced by the flow duration curve.

Table 1: consider indicating how much of the time there is zero flow.

Reply: This table has been replaced by the flow duration curve. We have added description of the time there is zero flow in text. It occupies only a short period of time in our observation period.

Table 1: why not provide a flow duration curve instead. That will we WAY more informative than what you currently present.

Reply: The flow duration curve has been added.

Line 159: CalculationS

Reply: We have revised this.

Line 162: for each of the (not in each of the)

Reply: We have revised this.

Line 162-163: inconsistent with singular and plural. Check grammar.

Reply: We have revised this sentence and made corrections in the whole manuscript.

Line 168: fix superscript "rainfall (m3 hour-1)" Equations 8-11: I presume you talk about some mean age for the box, please specify this. Equations 8-11 there equations are missing the "aging" term. (i.e. water gets older over time), please add this term and check if you calculations are correct: : :

Reply: We have revised this, and given a more complete description in the appendix.

We have considered the "aging" item. In the model procedure, each age item at the time t includes the age at the previous time step t-1. So, the results listed in this paper include the "aging effect" (this has been clarified in the appendix about the model descriptions).

Section 3.2 months spin up time may be sufficient spin up time for hydrometric fluxes, but will it be for modeling of ages?

Reply: The spin up time for the modeling of ages is sufficient given the young water dominance. Our two step calibration procedures show: as the mean value of the modelled water ages (meeting the target of KGE>0.3) in the first calibration was used as the initial water ages for the second calibration, the calibrated water ages for each conceptual store well matches the measured isotope values (the target of KGE increases to be higher than 0.5). It means that the selected initial period for "warm up" modeling of ages is reasonable.

Section 3.2: "First, different parameter combinations within the initial ranges in Table 3 were tested. And then, the parameter ranges were reduced according to the best models (KGE >0.3) for the second calibration. This resulted in a total of 10ˆ5 tested different parameter combinations. I do not understand how you arrive at the second 10ˆ5.

Reply: We have revised the descriptions and added the initial parameters from the first calibration (see lines 251-257 in the revised manuscript). A total of 10ˆ5 different parameter combinations was given for the estimation of uncertainty of the modeled results from the random generation of the possible parameter combinations. The number of 10ˆ5 different parameter combinations is believed to be sufficient according to uncertainty analysis in the literatures (Soulsby et al., 2015; Xie et al., 2017). In the first and second calibrations, the number of parameter combinations were set to 10ˆ5, but the range of the initial parameters are different (Initial range 1 and 2 for the 1st and 2nd calibration in revised Table 2).

After the first calibration when KGE >0.3, the range of each parameter was reduced. Then, the narrowed ranges (initial range 2 in Table 2) were used as the initial ranges for the second calibration.

**Table 2 Mean parameter values and fitness derived from the best 500 parameter sets after calibration**

| For Flow | $K_s$ (hour$^{-1}$) | $K_f$ (hour$^{-1}$) | $K_e$ (hour$^{-1}$) | $f$ | $a$ | $W$ | $b$ |
|---|---|---|---|---|---|---|---|
| Initial range 1 | 40-168 | 1-72 | 800-2200 | 0.005-0.025 | 0-1 | 0-0.015 | 0-1 |
| Initial range 2 | 40-150 | 1-40 | 800-2200 | 0.008-0.025 | 0.47-1 | 0-0.015 | 0.48-1 |
| Mean | 92 | 11 | 1549 | 0.015 | 0.68 | 0.005 | 0.54 |
| Range | 48-120 | 5-18 | 1000-2000 | 0.01-0.02 | 0.51-0.9 | 0.003-0.01 | 0.5-0.62 |
| For Isotope | *Is* | *KK* ($\times 10^4$) | *pp* | *con* | *fei* | Index | Mean(range) |
| | | | | | | $KGE_d$ | 0.85 (0.81-0.87) |

| | Initial range 1 | 0-1 | 0.8-1.6 | 0-1 | 0-1 | 0-1 | KGE$_i$ | 0.56 (0.52-0.59) |
|---|---|---|---|---|---|---|---|---|
| Initial range 2 | 0-0.8 | 0.8-1.6 | 0-1 | 0-1 | 0.5-1 | KGE | 0.7 (0.72-0.66) |
| Mean | 0.24 | 1.26 | 0.49 | 0.56 | 0.82 | | |
| Range | 0.002-0.6 | 1-1.5 | 0.02-0.95 | 0.04-0.97 | 0.71-0.93 | | |

Line 276: "rogue" ?   what do you mean

Reply: It refers to some samples that are unusually high during the study period (in Fig.4b). These samples could be affected by the paddy water in the manuscript description.

Figure 10: these values cannot be correct since the areas under these curves do not add up to 1.

Reply: The figure gave the probability density functions (PDFs) of the flux ages from the three units. The sum of these values for the three units, e.g. the total areas covered by the three curves with different colors (conceptual stores), equals to 1.

Line 420: cannot instead of can't.

Reply: We have revised this.

Line 445-447 "Given the results on water storage dynamics and the relative contribution to the fast flow reservoir shown in Figures 7 and 8, it can be deduced that the storage change within each conceptual store is the main driver of hydrological connectivity between them." Is this not just how you defined that the catchments functions yourself? So what did we really learn in the end? (also remove the "s" in stores)

Reply: We have revised the conclusions. The tracer-aided model supports general appropriateness of the model structure which related connectivity dynamics to storage change within different landscape units. During the dry period, there is weak hydrological connectivity between the hillslope and depression due to low storage. In contrast, during the wet period, hydrological connectivity between the hillslope and depression strengthens as water storage increases. In the early recession, after heavy rain, large fractures in the hillslope fill, leading to large water fluxes into the depression. Then, as storage declines, fluxes decrease and the hydrological connectivity weakens (lines 457-462 in the revised manuscript).

**Anonymous Referee #2**

**Main comments**
1. The study is well-written and concise. The model calibration and sensitivity analysis are detailed and well described However, there are some remaining concerns that the authors may account for before their manuscript can be considered for publication:
We sincerely thank the reviewer for carefully reviewing our manuscript and for the thoughtful, constructive feedback.

2. For the reader, who is not familiar with the author's preceding work, it is not clear how the model works. The schematic description in Fig 2 indicates that ET is taking place from the slow and fast karst groundwater storages, which would be quite unusual. To avoid misconception, please provide a complete model description in appendix (the table A1 is hardly understandable).

Reply: In this model, the karst critical zone in the hillslope was conceptualized as one reservoir, but the water stored in the reservoir was further sub-divided into upper active and lower passive storage zones (Fig 3 in the revised manuscript) for the simulation of isotope ratios and estimation of water ages. This division follows our previous measurements of the vertical distribution of the rock fractures/conduits along hillslopes where the large rock fractures/conduits decrease exponentially in the vertical direction (Zhang et al., 2011).

The karst critical zone in the depression was conceptualized as two connected reservoirs, fast and slow flow, representing the solutional conduits in karst aquifers connecting with intergranular pores and fractures (often termed as matrix porosity).

The evapotranspiration could occur from the rich conduit/fracture areas by extended plant roots in the deep aquifer (Rong et al., 2011). Therefore, evapotranspiration is sourced from both the fast and slow reservoirs in the model.

We have revised the model descriptions in an appendix for clearer explanation of the module functions and meanings in the revised manuscript.

3. Some clarification on where the novelties of this work start is necessary. The authors inform the reader that in Zhang et al. (2017), the model was developed in previous work that used tracer data in addition to stream discharge to constrain the model structure, improve parameterization, and aid calibration. If this was done before, and the methods only describe how the isotope enabled model was parametrized and evaluated, what is the novelty of this particular study?

Reply: The model in the preceding work (Zhang et al., 2017), conceptualized the flow and the geochemical solute (Ca+Mg) routings using conceptualization of the dual flow system at the catchment scale. So, the original model had no basis for disaggregating the hydrological connectivity between different landscape units (e.g. "hillslope- to- depression- to- stream" in the study catchment). The hillslope-depression is a typical landform with variable hydrological connectivity in the karst catchments in southwest of China (Figure r1, Chen et al., 2018). Here, we

improved our previous model structure by conceptualizing the hillslope and depression units (the improved part is in the red dotted box in Figure r2), and then use the hourly discharge and isotope values to calibrate the model. In addition, the new model has the parameters to represent passive storage inferred by isotope damping and the function of estimating the water ages from various landscape units in the catchment.

Although the tracer-aided model enhanced our understanding of the hydrological connectivity between different landscape units and the mixing processes, it increased the model parameters in the tracer modules. Therefore, we also evaluated the uncertainty of the simulation results including flow discharges, isotopic values, storages and ages at the different landscape units in this study.

[Figure]

Figure r1 Sketch map of karst hydrological processes (Chen et al, 2018)

[Figure]

Figure r2 Structure of the improved model, and the improved part is in the red dotted box

4. Figure 4 shows that only 5 of 12 parameters are sensitive, which is quite a low number. Usually, discharge contains enough information to identify 4-6 parameters (Jakeman & Hornberger, 1993). Adding of additional information like isotopes should increase this number, if the model structure is well-chosen. To check the contribution of discharge data and isotopes, could the authors show the parameter sensitivities using discharge or water isotopes only?

Reply: The trace-aided model includes 12 parameters, seven for flow routing (*Ks, Kf, Ke, f, a, w,* and *b*) and five for isotope ratios and water ages *(Is, KK, pp, con* and *fei)*. So, the overall model increased by five parameters in the isotopic module.

We analyzed the parameter sensitivities using either the outlet discharge and/or water isotopes. Targeting the discharge, six parameters (except w) among the seven parameters in the flow routing module are sensitive and the parameters in the isotopic module are all insensitive (Fig r3 (a)). Targeting only isotopic values and both flow discharge and isotopic composition, the sensitive parameters are same, including *Kf, a,* and *b* in the flow routing module, and *Is* and *fei* in the tracer module) (Fig r3 (b) and (c)). Using both flow discharge and isotopic composition as the target, these parameters were more sensitive than those using only isotopic values (see the wide ranges of the cumulative distributions in Fig r3 (c)).

Interestingly, increasing the two sensitive parameters in the isotopic module (the coefficient for evaporation fractionation *Is* and the weighted isotope composition of rainfall input by the parameter *fei*) results in three parameters in the flow module becoming insensitive (slow reservoir constant (*Kf*), the exchange constant between the two reservoirs *Ke* and the ratio of porosity of the quick to slow flow reservoir *f*).

This can be explained as follows: the former two sensitive parameters in the isotopic module emphasize atmospheric effects on the outlet flow (being "old/new"). Larger *Is* indicates more evaporative effect on the stored water, leading to the stored and released water being older, particularly during the dry period. Larger *fei* indicates newer rainfall recharge (more negative isotopic values) into aquifer, leading to the stored and released water being newer during rainfall period. Alternatively, the latter three parameters in the flow module emphasize effects of fast (newer) and slow (older) flows in aquifer on the outlet flow (being "old/new"). More water release from the slow reservoir (larger *Kf*) and greater release of the slow reservoir into the fast reservoir (larger *Ke*) could lead to the released water being older in the dry season; a high proportion of the fast flow storage (larger *f*) and a greater exchange between the fast reservoir and the slow reservoir (larger *Ke*) could lead to the released water being newer in the wet season.

Consequently, there is equifinality for these parameters in the trace-aided model, which can be overcome only when we have additional data to constrain some of the parameters, e.g. knowing the evaporative effect on water *Is* and the weighted isotope composition of rainfall input by the parameter *fei*.

We have added the above reasoning in our discussion of the revised manuscript.

[Figure]

(a) Sensitive parameters include *Ks, Kf, Ke, f, a,* and *b*

[Figure]

(b) Sensitive parameters include *Kf, a, b, Is* and *fei*

[Figure]

(c) Sensitive parameters include $K_f$, $a$, $b$, $Is$ and $fei$

Figure r3 Sensitivity of 12 model parameters using (a) flow, (b) isotope composition and (c) combined simulation of flow and isotopic composition. (The parameters inside the gray dotted box are for flow routing, and the outside parameters are for isotope routing.)

5. With a large fraction of the model parameters insensitive, how conclusive are the interpretations on the model internal dynamics that the authors use to explain connectivity and water age distribution in the system? In some of the figures, uncertainty ranges are provided and they are quite wide. In other figures (e.g., Fig. 5), only the mean is provided although the parameters controlling the observed processes ("w" in case of Fig. 5) are insensitive.

Reply: In the revised manuscript, we have described the uncertainty of the modeled results for the various landscape units in the catchment. We reached the following conclusions:

The outlet hydrometric and isotope observations (consisting of mostly young and fast flows) were used as the calibration targets in this study. The outlet simulations had the least uncertainty, while uncertainty in the hillslope and depression units were highly related to their hydrological connectivity with the outlet. The simulated fast flows in the hillslope and depression units had lower uncertainty than the simulated slow flows in the depression since the two former units are highly connect with the outlet.

Although some parameters (e.g. $w$ controlling hillslope flow dynamics) are insensitive, uncertainty bands of the hillslope flow (Fig r4) are narrow and the model captures quite well the hillslope

seasonality and event-based dynamics through targeting the best matching of outlet discharges and isotopic values. This indicates that the hillslope dynamics are closely linked to the outlet dynamic patterns (with strong connectivity between them), which is consistent with the ranges of δD and δ¹⁸O values at the hillslope spring being close to the ranges at the outlet discharge in Table 1 in the revised manuscript. (see lines 328-331 in the revised manuscript)

We discussed the modelled uncertainty in the discussions (see Section 5.4 in the revised manuscript)

[Figure]

Figure r4 Observed discharge at hillslope spring against the simulated discharge of hillslope unit (values are normalized)

**Detailed comments**

Line 66: There are a few studies on water storage, flux and age dynamics using tracers in karst environments.
Reply: we have revised this expression.

Line 132: If this was done before, what is the novelty of this study?
Reply: Please see response to Q3 in the main comments above.

Line 145: Is there are distinction between soil/epikarst and groundwater? What controls matrix-conduit exchange?
Reply: We have revised the descriptions. In this model, we conceptualized the groundwater aquifer in the depression by a dual flow system (involving fast and slow flow reservoirs), and the groundwater aquifer in the hillslope by an upper active storage (mostly from epikarst) mixing with a lower passive storage since the rock fractures/conducts reduce with depth from the ground surface in the hillslope profile according to our previous investigations.

The exchange between matrix and conduit is controlled by the water storage (relate to water level)

and the exchange constant between the two reservoirs (*Ke*) in each reservoir.

Line 222: This is not correct - please remove
Reply: We have revised this.

Line 260: typo
Reply: We have revised this.

Line 296: only 5 of 12 parameters are sensitive, which is quite a low number. Usually, discharge contains enough information to identify 4-6 parameters (Jakeman & Hornberger, 1993). Adding of additional information like isotopes should increase this number, if the model structure is well-chosen. To check this, could the authors show the parameter sensitivities using discharge only?
Reply: Please see the reply to Q4 in the main comments above.

Line 299: In the text, a rejection limit of 0.3 is mentioned. Please clarify
Reply: Two step calibrations were carried out in this study. First, $10^5$ different parameter combinations were selected with the broad ranges of initial parameter values. And then, we obtained the narrower ranges of the parameters according to the best modelled results (meeting the KGE >0.3 criteria). For the second calibration, the narrowed ranges of the parameters were used as the initial ranges of the parameter to search the next best modelled results ( KEG >0.5). We have revised the descriptions in the new manuscript accordingly.

Line 308: The parameter w is completely insensitive meaning that this storage's dynamics are not well identifiable, right? Please provide all behaviorals instead of the mean to show the precision of simulation of its discharge
Reply: Please see the reply to Q5 in main comments above.

Line 324: how much can you conclude from such wide uncertainty ranges?
Reply: The greater uncertainty of the modelled isotopic values in the depression arose from the insensitive parameters of *Ke* and *Ks* that affect the slow flow discharge and its exchange with the fast flow when the outlet hydrometric and isotope observations (consisting of mostly young and fast flows) used as metrics for the objective function for model calibration.

Here, the modelled isotope composition in the depression (see Figure 6b) refers to the release of water from the slow flow reservoir, representing a relatively constant source. The uncertainty bands can cover the limited variability of the measured values of δD at W1 and W5 (blue and yellow points in Figure 6b) where the aquifer has much lower permeability (W5) and is confined (W1) (cf the geophysical survey reported by Chen et al, 2018). This means that our tracer-aided model captures the slow flow dynamics in the depression even though the uncertainty is large.

The highly negative values of δD at W3 and W4 (red and black points in Figure 6b) are mostly below the uncertainty bands. This means that the stored water at W3 and W4 was younger than water from the slow flow reservoir, which is consistent with recent geophysical evidence (see Chen et al, 2018). Since W3 and W4 are located at high permeability areas, water at W3 and W4 was contributed mostly

by fast flows (mixing with the young water), particularly during rainfall events (e.g. 9/7, and 20/7 in Fig 6b). So the high negative values of δD at W3 and W4 below the uncertainty bands were reasonable. We have revised the descriptions accordingly in the new manuscript. (see lines 344-351 in the revised manuscript)

Line 368: KE is also quite insensitive. Can you also show the entire 500 ensamble (or confidence limits)?
Reply: The simulations from the entire 500 ensemble are shown in Fig r5 (or Fig 9 in the revised manuscript). Since the parameter of *Ke* that determines the exchange amount between the fast and slow flow reservoirs is insensitive, the simulated exchange flux is highly uncertainty, though much smaller, compared to the water fluxes from the rainfall recharge and hillslope flow.

[Figure]

Figure r5 Source contributions to the underground stream flow (fast reservoir) at the catchment outlet. The red dots above and under the dotted line represent transient reverse water fluxes from the slow reservoir to fast reservoir and fast reservoir to slow reservoir, respectively.

Line 387: Please double-check this with literature values. Fast flow components in karst systems provide water with ages mostly between days or weeks (including temporal storage in the epikarst). Mostly, the ages found here are too large, even in the wet period.

Reply: We believe that the estimated ages are reasonable. Most models do not include mixing processes with stored water so tend to under-estimate water ages.

Here we listed δD values at the sampling points in this catchment for the two largest rainfall events

in 2017 (the details refer to Chen et al., 2018, https://doi.org/10.1002/hyp.13232).

| Date | rainfall amount | rain water | outlet water | hillslope spring |
|------|-----------------|------------|--------------|------------------|
| 12/6 | 86.6 mm | -85 | -48 ~ -70 | -62~-67 |
| 9/7 | 83.4mm | -80 | -62~ -73 | -59~ -70 |

It shows that δD values at outlet and hillslope spring are much less negative than rainwater. So there was strong mixing of the "new" rainwater with "old" stored water during and after the rainfall although the response of discharge to rainfall is fast.

Also, our estimated ages in the manuscript refer to the mean of the ages over a long period of time. For short-term (event based) responses to the rainfall, the ages of water from hillslope flow and fast reservoirs can be shortest as 4 and 2 days, respectively. There were 8 and 23 events for the fast flow with the ages of water less than 5 and 10 days, respectively (see the lowest values in Fig 9). So, the results are not inconsistent with previous work, rather capture the time-variance of water ages. We have added these explanations in the revised manuscript (see lines 412-414 in the revised manuscript).

Line 398: See comment above
Reply: The same response as for Line 387.

Line 409: Some recent example how this can be done with water quality data in karst:
Hartmann, A., Barberá, J. A., & Andreo, B. (2017). On the value of water quality data and informative flow states in karst modelling. Hydrology and Earth System Sciences, 21, 5971–5985.
https://doi.org/10.5194/hess-2017-230
Reply: We have added this relevant literature.

Line 441: Large fractions of the fast reservoir have ages larger than several months, which appears a bit slow. (see also comments above)
Reply: Please see response to Line 387.

**The list of all relevant changes in the revised manuscript**

1. Some grammatical errors in the manuscript, related to the use of articles and plural/singular, have been corrected;
2. We have adjusted the Table and Figure numbers, and then revised these in the new manuscript accordingly;
3. We have deleted the Table 1 in original version and added a flow duration curve, according to the referee's suggestion;
4. Line 14-12 and Line 17-27: We have rephrased the abstract;
5. Line 52-58: We have revised this paragraph;
6. Line 61-65: We have revised these sentences;
7. Line 101: We have added the relevant literature;
8. Line 118-119: We have added the relevant descriptions;
9. Line 125-128: We have added the description of the time there is zero flow;
10. Line 130: We have added the flow duration curve;
11. Line 144-154: We have rephrased the Modelling approaches;
12. Line 177-179 and Line 193-194: We have added the equation descriptions;
13. Line 198: We have added the description of partial mixing;
14. Line 201-202 and Line 209-210: We have added the equation descriptions;
15. Line 217-218: We have added the description of the "aging effect";
16. Line 227-229: We have added the description of parameters;
17. Line 251-257: We have revised this paragraph;
18. Line 289: We have revised Fig.4b;
19. Line 316: We have revised Table 2;
20. Line 328-331: We have added the description of the uncertainty of the modeled results;
21. Line 332: The original figure have been replaced by a revised plot;
22. Line 344-351: We have revised the descriptions of the uncertainty ranges;
23. Line 369-373: We have revised this paragraph;
24. Line 399-401: We have revised this paragraph;
25. Line 402: The original figure have been replaced by a revised plot;
26. Line 412-414: We have added the description of water age;
27. Line 423-428: We have revised the descriptions of the uncertainty ranges;
28. Line 449-474: We have rephrased this paragraph about the hydrological connectivity.
29. Line 518-547: We have added the discussion of the equifinality of model parameters and uncertainty of the modelled results;
30. Line 565-568: We have revised the conclusions;
31. Line 586-587: We thanks the two anonymous reviewers and the editor for their constructive comments;
32. Line 641-642: We added the reference;
33. Line 652-653: We added the reference;
34. Line 713-714: We added the reference;
35. Line 743-746: We added the references;
36. Line 755: The Table A1 in Appendix have been replaced by a revised table;
37. We have added the supplementary material.

[revised manuscript text omitted]

---

## Referee Report (RR1)

Review Hess-2018-205 version 2

Storage dynamics, hydrological connectivity and flux ages in a karst catchment: conceptual modelling using stable isotopes

By Zhang et al.

The paper describes a detailed and data-rich study on applying conceptual tracer-aided modelling to understand the hydrological response of a karstic catchment. It does so by also considering hillslopes besides the more classical slow and fast subsurface reservoirs. It is well written, structured and presented. Although the presented methods are not novel, their application in karstic environments has novelty and value to the hydrological community. The work has been done, as far as I can tell, in great detail and profoundness.

I have not been involved in the first round of reviews. Reading the discussion, the authors improved their article (mainly clarification and some more detailed discussion) based on the reviews. One remark, however, I must admit I am quite disappointed by the data policy statement that the data are not made openly available. This hinders scientific progress for no obvious reason, it also obstructs verification of the data and modelling. I hope the authors can do another attempt freeing the data. Notwithstanding this annoying shortcoming, I think the article can be published after taking into consideration the minor questions and technical correction listed below.

One point of concern relates to the mean age distribution, as already pointed out in the first round of reviews. Can the authors discuss what is the value of a mean age of the discharged water if the reservoir storages are so small: hillslope unit has V=23 mm (while being 0.88 of the 1.25 km2 surface area) and the fast reservoir has a mean storage volume of 0.2 mm (with max of 6-7 mm).

My second point relates to the groundwater sampling. L135: How did you sample the GW? Did you remove some pore volumes before taking the sample? What is exactly an isotope value over 13 or 35 m filter length (as screening is over entire length)? How does that influence your modelling results of the slow reservoir? Would sampling over smaller filter screens not be more informative?

Technical and minor remarks:

The use of symbols in the article are not according to HESS standards. HESS promotes use of single symbol representation for parameters, variables etc. (please replace con-fei-KK-Age etc with single letter symbols

L248: the all > all the

Fig 4: lc-excess panel impossible to read. Maybe enlarge (or add enlarged version in supplement?)

Fig 4: panel a: typo in legend. Aimulations > simulations

 L362: why can the storage volume of 0.2mm of the fast reservoir be explained with "because the underground river/conduit volume represents only a very small proportion of the porosity of the

entire aquifer". A huge conduit above the local groundwater level would have no storage in your model but take up quite some proportion of the subsurface, isn't it. Please explain

L391 (and on some more place). Why "unique". It is a situation we see more in e.g. large fissures/preferential flow paths in soil hydrology / subsurface hydrology, in urban area (mentioned later on in the article) etc. It is a normal situation that can be explained physically easily. Therefore, I do not see this as something "unique", but "distinct or characteristic"

Fig 9: negative sign missing in lower part plot

L459: is the low connectivity due to low storage or due to low hydraulic gradients (in unsaturated subsurface)?

L464: here I agree, the low storage is not the driver, the hydraulic gradients are the driver and storage can be a useful index for it in lumped conceptual modelling. I would suggest to say hydraulic gradients ((such as gradients in water levels are)

L468: also here: Unique? Replace by 'distinct/characteristic'

L472: Suggest to add preferential flow in large fissures, cracks in soil hydrology

 L488: "release younger water"? The fast system cannot hold on to the water, so how does it release it?

L491: Typo Jasecho

Thom Bogaard

Delft University of Technology

The Netherlands

---

## Author Response (AR2)

Review Hess-2018-205 version 2
Storage dynamics, hydrological connectivity and flux ages in a karst catchment: conceptual modelling using stable isotopes
By Zhang et al.

The paper describes a detailed and data-rich study on applying conceptual tracer-aided modelling to understand the hydrological response of a karstic catchment. It does so by also considering hillslopes besides the more classical slow and fast subsurface reservoirs. It is well written, structured and presented. Although the presented methods are not novel, their application in karstic environments has novelty and value to the hydrological community. The work has been done, as far as I can tell, in great detail and profoundness.

We sincerely thank the reviewer for his comments and suggestions. We have revised our manuscript taking into account all suggestions and comments from the reviewer. Our point-to-point responses are detailed below.

I have not been involved in the first round of reviews. Reading the discussion, the authors improved their article (mainly clarification and some more detailed discussion) based on the reviews. One remark, however, I must admit I am quite disappointed by the data policy statement that the data are not made openly available. This hinders scientific progress for no obvious reason, it also obstructs verification of the data and modelling. I hope the authors can do another attempt freeing the data. Notwithstanding this annoying shortcoming, I think the article can be published after taking into consideration the minor questions and technical correction listed below.

Reply: The data can be shared after ending of our project (2019) according to the project executive policy. Anyone who likes to use the data can contact the corresponding author after signing agreement.

One point of concern relates to the mean age distribution, as already pointed out in the first round of reviews. Can the authors discuss what is the value of a mean age of the discharged water if the reservoir storages are so small: hillslope unit has V=23 mm (while being 0.88 of the 1.25 km2 surface area) and the fast reservoir has a mean storage volume of 0.2 mm (with max of 6-7 mm).

Reply: The age in our study refers to the age of flux, which reflects the integration of the nonstationary water age distributions in the fluxes from the different sources. It can be tracked using a time stamp that tags each hourly incoming flux and outflowing flux via their movement through the storage cascade (Hrachowitz et al. 2013; Soulsby et al., 2015):

$$Age = (\sum_{n=1}^{N} Age_{Qin,n} \, Q_{in,n} + Age_{res} V_{res})/V$$

where $Age$ is the mean age of flux from conceptual store, $Q_{in,n}$ and $Age_{Qin,n}$ are the incoming water volume and corresponding water age, respectively, $V_{res}$ and $Age_{res}$ are resident water storage volume and corresponding water age, respectively, $V$ is the total storage volume of conceptual store at each end of time step. The water age of flux from the corresponding conceptual store is controlled by the relative proportion of water volume (incoming and resident) and their respective ages.

For the fast flow in depression, the mean age of the discharged water from fast reservoir generally

reflected the integration of the incoming water, including younger water fluxes from the hillslope and rainfall infiltration, and older fluxes from the slow flow reservoir, which mixes with the previous stored water. The smaller storage of fast reservoir reflects that it takes less regulation on the income water. During the rainfall, the age of water in fast reservoir (being the shortest of 2 days) indicates the fast-released fresh water from the rainfall infiltration and hillslope flow, which little mixes with the previous stored water due to the small storage and. In contrast, in the non-rainfall period, the age of water in fast reservoir (being the longest of 466 days) reflects slow released water mainly from slow flow reservoirs (the ages between the fast and slow flow reservoirs are close in Fig 10).

Similarly, for the hillslope flow, the mean age of the discharged water reflected the integration of the incoming water from rainfall infiltration, which mixes with the previous stored water as well as the inactive water (Fig 3). The storage of hillslope flow is larger than the storage of fast reservoir in depression due to rich fractures in epikarst zone but much less than that of slow flow reservoir. The relative smaller storage of hillslope flow takes a similar function on the age variations: fast release from rainfall during rainfall and slow release from the inactive layer in the non-rainfall period.

**My second point relates to the groundwater sampling. L135: How did you sample the GW? Did you remove some pore volumes before taking the sample? What is exactly an isotope value over 13 or 35 m filter length (as screening is over entire length)? How does that influence your modelling results of the slow reservoir? Would sampling over smaller filter screens not be more informative?**

Reply: We used a kind of special sampling instrument to collect the sample at a specific depth. At each well, water was sampled from multiple depths with a depth‑specific sampler to give a profile of the isotopic composition of the groundwater column (refer to Chen et al., 2018). Thus, the analysis results in Table 1 refers to mean value of the samplings at the different depths.

Here, water samplings at Well 1 and 5 represent water from the slow reservoir since they are located at the less permeable aquifer (Chen et al. 2018). Our simulated isotope values of the slow flow cover the isotope variation from the samplings (see the dotted $\delta D$ values at these two wells within the uncertainty bands in Fig. 7 in the manuscript). In contrast, the isotope values from the water samplings at Wells 3 and 4 represent those from the fast flow reservoir. Thus, $\delta D$ values at Wells 3 and 4 were underlined the uncertainty bands in Fig. 7 (meaning younger than water at the slow reservoir).

Due to high heterogeneity of the karst critical zone, it is difficult to separate "slow" and "fast" components at any wells, which may result in the greater uncertainty in Fig 7 from the modeling results.

**Technical and minor remarks:**

**The use of symbols in the article are not according to HESS standards. HESS promotes use of single symbol representation for parameters, variables etc. (please replace con-fei-KK-Age etc with single letter symbols**
Reply: we have revised this.

L248: the all > all the

Reply: we have revised this.

Fig 4: lc-excess panel impossible to read. Maybe enlarge (or add enlarged version in supplement?)
Reply: we have revised this.

Fig 4: panel a: typo in legend. Aimulations > simulations
Reply: we have revised this.

L362: why can the storage volume of 0.2mm of the fast reservoir be explained with "because the underground river/conduit volume represents only a very small proportion of the porosity of the entire aquifer". A huge conduit above the local groundwater level would have no storage in your model but take up quite some proportion of the subsurface, isn't it. Please explain
Reply: It is a typical karst hydrogeological feature: fast flow and small storage capacity for the conduits and underground channels (Birk, et al., 2004; Einsiedl, 2005).
In the whole catchment, the conduits and underground channels occupy a small proportion of the catchment volume. According to field surveys in other karst catchments, the mean global porosity of karst conduit networks is less than 1% of the catchments (Pardo-Iguzquiza et al., 2011; Bonacci et al., 2006). In our conceptual model, the numerical conduits above the local groundwater level were treated as the fast flow reservoir with a small storage. The fast flow reservoir takes a function of fast recharge into the slow reservoir during the rainfall and fast release of the subsurface flow after rainfall.

Birk, S., Liedl, R. and Sauter, M.: Identification of localised recharge and conduit flow by combined analysis of hydraulic and physico-chemical spring responses (Urenbrunnen, SW-Germany), J. Hydrol., doi:10.1016/j.jhydrol.2003.09.007, 2004.
Einsiedl, F.: Flow system dynamics and water storage of a fissured-porous karst aquifer characterized by artificial and environmental tracers, J. Hydrol., doi:10.1016/j.jhydrol.2005.03.031, 2005.
Pardo-Iguzquiza, E., Durán-Valsero, J. J. and Rodríguez-Galiano, V.: Morphometric analysis of three-dimensional networks of karst conduits, Geomorphology, doi:10.1016/j.geomorph.2011.04.030, 2011.
Bonacci, O., Ljubenkov, I. and Roje-Bonacci, T.: Karst flash floods: An example from the Dinaric karst (Croatia), Nat. Hazards Earth Syst. Sci., doi:10.5194/nhess-6-195-2006, 2006.

L391 (and on some more place). Why "unique". It is a situation we see more in e.g. large fissures/preferential flow paths in soil hydrology / subsurface hydrology, in urban area (mentioned later on in the article) etc. It is a normal situation that can be explained physically easily. Therefore, I do not see this as something "unique", but "distinct or characteristic"
Reply: we have revised this.

Fig 9: negative sign missing in lower part plot
Reply: we have revised this.

L459: is the low connectivity due to low storage or due to low hydraulic gradients (in unsaturated subsurface)?
Reply: We changed it to be "due to the low hydraulic gradients and/or hydraulic conductivity".

L464: here I agree, the low storage is not the driver, the hydraulic gradients are the driver and storage can be a

useful index for it in lumped conceptual modelling. I would suggest to say hydraulic gradients ((such as gradients in water levels are)

Reply: We changed the sentence to "The hydrological connectivity and exchange between the slow and fast flow reservoirs is mainly controlled by the hydraulic gradients between two mediums, rather than the storage."

L468: also here: Unique? Replace by 'distinct/characteristic'

Reply: we have revised this.

L472: Suggest to add preferential flow in large fissures, cracks in soil hydrology

Reply: we have added the statement of preferential flow in cracks in soil.
"In this regard, it is similar to the cracks in soil leading to high percolation via preferential flow paths under flooding condition (Zhang et al, 2014)."

L488: "release younger water"? The fast system cannot hold on to the water, so how does it release it?

Reply: we change expression, the "release" was replaced by "transit".

L491: Typo Jasecho

Reply: we have revised this.

**The list of all relevant changes in the revised manuscript**

1. Some symbols which are not according to HESS standards in the article have been corrected;
2. Line 248: We have revised the word;
3. Fig.4: The original figure has been replaced by a revised plot;
4. Line 391: 'unique' has been replaced by 'characteristic'
5. Fig.9: The original figure has been replaced by a revised plot;
6. Line 459: We have revised the sentence;
7. Line 464: We have revised the sentence;
8. Line 468: 'unique' has been replaced by 'distinct';
9. Line 472: We have added the relevant descriptions;
10. Line 488: We have changed the expression;
11. Line 491: We have the typo;
12. We have changed the *Data availability*;
13. We thank Thom Bogaard for his constructive comments in *Acknowledgments*;
14. The original figures in supplementary material have been replaced by the revised plot.

[revised manuscript text omitted]